

# Glaciation's topographic control on Holocene erosion at the eastern edge of the Alps

Jean L. Dixon[1,2], Friedhelm von Blanckenburg[1,3], Kurt Stüwe[4], and Marcus Christl[5]

[1] German Research Centre for Geosciences GFZ, Potsdam, Germany
[2] Department of Earth Sciences and the Institute on Ecosystems, Montana State University, USA
[3] Department of Geosciences, Freie Universität Berlin, Germany
[4] Institute of Earth Sciences, University of Graz, Austria
[5] Laboratory for Ion Beam Physics, ETH Zürich, Switzerland

*Correspondence to*: Jean L. Dixon (jean.dixon@montana.edu)

**Abstract.**

What is the influence of glacial processes in driving erosion and uplift across the European Alps? It has largely been argued that repeated erosion through glaciation sustains isostatic uplift and topography in a decaying orogen. But, some insist that the Alps are an orogen still actively uplifting (e.g., Hergarten et al., 2010). We add insight to this debate by isolating the role of post-glacial topographic forcing on erosion rates. To do this, we quantify the topographic signature of past glaciation on millennial scale erosion rates in previously glaciated and unglaciated catchments at the easternmost edge of the Austrian Alps. Newly measured catchment-wide erosion rates, determined from cosmogenic $^{10}$Be in river-borne quartz, correlate with basin relief and mean slope. GIS-derived slope-elevation and slope-area distributions across catchments provide clear topographic indicators of the degree of glacial preconditioning, which further correlates with erosion rates. Erosion rates in the eastern-most, non-glaciated basins range from 40 to 150 mm/ky and likely reflect underlying tectonic forcings in this region, which have previously been attributed to recent (post 5 Ma) uplift (Legrain et al., 2015). By contrast, erosion rates in previously glaciated catchments range from 170 to 240 mm/ky and reflect the erosional response to local topographic preconditioning by repeated glaciations. Together, these data suggest that Holocene erosion across the Eastern Alps is strongly shaped by the local topography relict from previous glaciations. Broader, landscape-wide forcings, such as the widely debated deep mantle-driven or isostatically-driven uplift, result in lesser controls on both topography and erosion rates in this region. Comparing our data to previously published erosion rates across the Alps, we show that post-glacial erosion rates vary across more than two orders of magnitude with poor topographic indicators of controls. This high variation in post-glacial erosion may reflect combined effects of direct tectonic and modern climatic forcings, but is strongly overprinted by past glacial climate and its topographic legacy.



# 1 Introduction

The climatic control on erosion in mountain belts remains a longstanding and active debate in geomorphology. Some of this debate has focused on whether spatial gradients in precipitation can be invoked to drive gradients in erosion or whether these rates are more strongly controlled by their tectonic setting (e.g., Burbank et al., 2003). While some studies have argued for
modern precipitation controls on erosion (e.g. Bookhagen et al., 2005), climate's imprints via glacial processes are widely recognized to significantly alter a landscape. For example, abrasion and plucking of bedrock by overlying glacial ice widens and deepens valleys (Brocklehurst and Whipple, 2002). Via this erosional 'buzzsaw', glaciers have been suggested to set the limit on mountain range height and relief (e.g., Egholm et al., 2009, Mitchell and Humphries, 2015) and accelerate mountain erosion (e.g., Herman et al., 2013). The isostatic rebound from glacial erosion and retreat causes uplift of rocks and increases
mountain relief (e.g., Champagnac et al., 2007; Molnar and England, 1990). Post-glacially, rivers export unconsolidated sediments stored in basins (Hinderer et al., 2001, Hoffmann et al., 2007), and steep glacial headwalls and valley sides undergo accelerated hillslope erosion. The resulting post-glacial sediments can become effective tools for rivers to rapidly incise their beds (Jansen et al., 2011). Glaciers also leave a lasting topographic legacy that influences erosion, relief, and possibly uplift tens of thousands of years after glacial retreat (e.g., Salcher et al., 2014). Together, these processes and observations may
suggest that glacial forcings are the dominant control on landscape evolution in mountain belts.

Notwithstanding the clear topographic and erosional effects that glacial processes imprint in the landscape, there has been notable pushback on the idea that climate via glaciation is the dominant driver of erosion in diverse mountain belts. For example, all glaciers are not efficient eroders, and glaciers frozen to their base may instead protect bedrock from erosion in
high topography (Thomson et al., 2010). Even across glacial-interglacial time periods, fluvial incision may outpace glacial erosion in valley bottoms (Montgomery and Korup, 2011). Furthermore, global compilations of erosion rates across multiple temporal scales show similar erosion rates by glaciers and rivers, and these data suggest that tectonics likely controls erosion rates over millennial and longer timescales regardless of glacial history (Koppes and Montgomery, 2009).

This debate on climate's influence has been especially active for the European Alps, where both glacial and tectonic forces have been invoked as principle drivers of erosion and uplift (Cederbom et al., 2004). Wittmann et al. (2007) noted that erosion rates exceed uplift rates in the central Alps, and that correlations between topography, uplift and erosion suggest glacial and post-glacial erosion alone may explain rates of uplift in the region via isostasy. Norton et al. (2010b) and (2011) further argued that glaciation drives uplift, based on the observation that youthful tectonic features such as river knickpoints are highly
correlated with previous glacial cover and glacial equilibrium line altitudes. However, it has also been suggested that ongoing collision and active convergence in the eastern Alps may either alone drive uplift (Hergarten et al., 2010) or significantly contribute to changes in relief across the Cenozoic (Legrain et al., 2014). In the eastern portion of the range, accelerated rates of river incision and hillslope erosion since 5 Ma have been suggested to record late Tertiary uplift resulting from deep



lithospheric processes (Legrain et al., 2015; Wagner et al., 2010). These relatively local observations have been coupled with landscape evolution models to suggest the Alps as a whole are not a decaying orogen as a glacial-driver of uplift and erosion may suggest, but instead a young mountain range still experiencing tectonic rejuvenation (Hergarten et al., 2010, Robl et al., 2015).

Here, we add insight into to the debate on the role of glaciers in driving Alpine erosion, by quantifying landscape morphology and [10]Be-derived denudation rates (hereafter called erosion rates) in both unglaciated and previously glaciated basins of the far Eastern Alps. We find that the past glacial history exerts a stronger control on erosion rates across the eastern Alps than previously invoked tectonic forcings.

## 10  2 Approach

### 2.1 Study Site

Our study region lies in the easternmost section of the European Alps, composing the Styrian as well as several intramontane basins and adjacent massifs that make up the Alpine uplands: The Lavanttal Alps (including Gleinalpe and Koralpe), the Schladming Tauern, the Sekauer Tauern, and Pohorje in Slovenia (Fig. 1). The Styrian Basin (part of the Pannonian Basin)
was a shallow marine basin throughout much of the Miocene, becoming brackish and finally freshwater during basin inversion, which commenced around 10 Ma (Bada et al., 1999; Cloetingh et al., 2006). These kilometer thick Miocene sediments now underlie a gentle hilly terrain that has uplifted some 300 m above sea level in the last 7 Ma. The upland regions of adjacent Massifs are made up of high grade metamorphic rocks, with local limestone in the range north of the basin.

During the glaciation periods of the past million years, only the western portion of the study region was pervasively glaciated (Figure 1). East of the contiguous Alpine ice cap, only isolated cirque glaciers occurred at elevations above 2000 m, for example in the summit region of the Koralpe range. In unglaciated portions of our study area, previous geomorphic work has recognized two distinct landscape morphologies: a low-gradient, low-relief upland region and a higher-gradient, higher-relief region downstream of river knickpoints (Legrain et al., 2014; Robl et al., 2008). Millennial erosion rates from small basins
within these regions correlate with slope and the degree of incision (Legrain et al., 2015). These two morphologies are interpreted to represent the relict and incising portions of a landscape responding to a propagating wave of incision initiated at ~4 Ma (Wagner et al., 2010). This timing coincides with inversion and uplift of the Styrian and northern Molasse Basins, but appears conspicuously unrelated. No work thus far has compared erosion rates in the previously glaciated and unglaciated portions of this landscape.



## 2.2 Deriving erosion rates from *in-situ* produced cosmogenic [10]Be

Use of the cosmogenic nuclide [10]Be in river sand is now standard for quantifying rates of erosion in diverse landscapes (Granger and Schaller, 2014; Portenga and Bierman, 2011; von Blanckenburg, 2005). Cosmic ray bombardment of Earth's surface produces these nuclides in-situ, and their concentrations reflect the time that minerals spend within the upper few meters of

Earth's surface. [10]Be concentrations in quartz collected from river sands reflect erosion rates spatially integrated across the basin. We sampled 26 rivers in the Eastern Alps of Austria and Slovenia for cosmogenic [10]Be analysis, targeting both previously glaciated and unglaciated catchments across the region (Table 1-2). Sand was collected from channel bottoms and active channel bars, integrating along ~20 m reaches at each river location. Samples were oven dried and sieved to extract the 250-500 um size fraction. Heavy and magnetic minerals were removed using magnetic and density separation methods.

Standard hydrochloric and hydrofluoric chemical leaches removed non-quartz minerals and etched weathering rinds from quartz to remove meteoric [10]Be. We digested 40 g of clean quartz in a 5:1 concentrated hydrofluoric acid: nitric acid mixture, along with 215 µg of a in-house developed [9]Be carrier derived from phenakite crystal. Beryllium was extracted from digested quartz and oxidized using methods outlined in von Blanckenburg et al. (1996). We measured [10]Be/[9]Be ratios on BeO targets with accelerator mass spectrometry at ETH Zürich in Switzerland in June of 2010 and 2011. Initial AMS results are normalized

to AMS standard S2007N, with an isotope ratio of 2.81 x 10[-11]. All results are renormalized to the 07KNSTD standardization. Table 1 presents analytical results. [10]Be concentrations are blank corrected by subtraction (average [10]Be/[9]Be ratio of five chemical processing blanks = 2.72 ± 2.21 x 10[-15]).

[10]Be concentrations were used to derive catchment-wide erosion rates, following scaling factors from Dunai (2000), absorption

laws for nucleonic interactions from Schaller et al. (2002), and muonic absorption laws from Braucher et al. (2003). We determined basin-averaged production rates using an ArcGIS-based production model, 10 m gridded elevation data, a sea-level, high latitude production rate of 4.0 atoms/$g_{qtz}$/year (Phillipps et al., 2016), and assuming slow and fast muons contribute ~1.2% and 0.65% of total production (Braucher et al., 2003) . Corrections for skyline shielding were made following Norton and Vanacker (2009). We calculated snow-shielding following Norton et al. (2008) using elevation-snow depth relationships

previously determined in the Swiss Alps by Auer (2003). Because our cosmogenic [10]Be concentrations only reflect erosion rates in the parts of the basin with quartz-bearing lithologies, we set production rates equal to zero in parts of drainage basins with carbonate terrains to calculate integrated basin [10]Be production rates (Table 1). [10]Be derived erosion rates are presented in Table 2.

## 2.3 Digital Terrain Analysis

Catchment topography was analyzed using two digital elevation models: 10 m gridded data available from the Austrian Geological Survey (BEV) and 80 m gridded data from the Global Shuttle Radar Topography Mission (SRTM). Terrain attributes, stream networks and catchment extents were extracted in ArcGIS on both sets of gridded data. Several catchments





lay within Slovenia, and outside the extent of the Austrian 10 m data. Table 2 provides basin-wide terrain attributes, including comparison of variables extracted from 80 m and 10 m digital elevation models. Though the scale of these data sets are very different, the resulting topographic metrics are quite similar, with only a slight lowering of average slopes in the coarser data. This similarity highlights the fact that local slopes are largely controlled by landscape-scale patterns. If local slopes were

5 variable at a small spatial scale, then analysis of 10 m and 80 m gridded data would result in notable differences.

## 3 Results and Discussion

### 3.1 Geographic distribution of denudation rates

Catchment wide erosion rates generally show distinct patterns based on their geographic regions across the Eastern Alps (Fig. 2a,b; Tables 1-2).   Rates across Gleinalpe and Koralpe range from 39-104 mm/ky. The erosion rates measured in catchments

entirely within the Styrian Basin (101-114 mm/ky; Fig. 1) are notably higher than the rates within the adjacent Koralpe range. Streams in these lowland-basin catchments drain largely unconsolidated sediments of Miocene age that form low-relief hillslopes. Tributaries of the Murz river valley in the northeast exhibit a broad range in erosion, from 81-151 mm/ky. Catchment erosion rates in the Schladminer and Seckauer Tauern range from 71-238 mm/ky. The highest rates in this region (>170 mm/ky) correspond to basins that lie within the range last glacial maximum ice and reflect the region that was previously glaciated (see

Fig. 1).

### 3.2 Correlations between denudation and topographic metrics

The broad regional differences in basin erosion rates are complemented by relationships between these rates and topographic form of the basins (Fig. 2a,b). Mean basin slope generally increases linearly with mean elevation (Fig. 3; $r^2$=0.64, p<0.001).

This increase in slope is partially controlled by a marked increase in the proportion of slopes that are steeper than 35° at high elevations (Fig. 3). Measured erosion rates also generally increase with increasing mean basin elevation ($r^2$=0.34, p<0.001) and slope (Fig. 2b; $r^2$=0.58, p<0.001). These data are consistent with trends previously observed across other mountain ranges (e.g., Cyr et al., 2010; Ouimet et al., 2009), whereby erosion rates increase non-linearly with mean catchment slope. This non-linear relationship may result either by the dominance of threshold driven landsliding in controlling erosion across the range

(e.g., Montgomery and Dietrich, 1994) or by non-linear dynamics in hillslope creep (Roering et al., 2001). Either of these erosional mechanisms may result in a similar form to the non-linear relationships between erosion rates and slope (e.g., DiBiase et al., 2010). We note that both erosion rates and catchment mean slope correlate with the proportion of the catchment that exceeds 35° (Fig. 2b), and that these steep slopes generally are void of soil cover.





Considering the overall trend of increasing erosion rates with basin slope and higher erosion rates in general, it is surprising that several basins at lowest elevations do not follow this trend (Fig. 2,4). Low elevation catchments in the Styrian basin to the south erode at faster rates than catchments in the middle uplands of the Koralpe range (Fig. 2). These high erosion rates at low elevation have previously been linked to tectonic transience in the Koralpe range, such that a wave of incision and erosion

propagating upslope has accelerated erosion, but not yet reached upper relict landscapes. Legrain et al. (2015) mapped the transition between incising and upland relict hillslopes, and found that erosion rates in small basins (< 1 km$^2$) across Koralpe correlate with the fraction of the catchment below transient propagating knickpoints. Catchment morphology and erosion rates within these small basins show greater variability at mid-to-low elevations than the larger basins studies here, and reflect the local topographic and erosional response of hillslopes to transient river incision (Legrain et al., 2014; Robl et al., 2008). Higher

rates in the Styrian Basin compared to uplands of Koralpe, therefore likely reflect this erosional response to river incision and tectonic processes across the range. However, with the exception of these high rates in the Styrian Basin, this local-scale topographic variability and transience attributed to tectonics is not strongly reflected in the large basins studied in this paper, which integrate across this variability spatially.

**3.3 Topography and erosion rates in previously glaciated and non-glaciated catchments**

Catchments in the Schladminger Tauern and northern parts of the Seckauer Tauern were glaciated in the Pleistocene (Fig. 2a). These catchments exhibit the most rapid erosion rates across the study area (Fig. 4a; 170 -230 mm/ky), and have higher average slopes than non-glaciated and only partly-glaciated basins (Fig. 4b,c,d). We hypothesize that these higher slopes in glaciated catchments reflect glacial sculpting of topography. However, basin average slope angles (Fig. 2) only provide limited proof of

concept since we find a wide range of mean values across unglaciated basins. Therefore, to better distinguish a topographic fingerprint of past glaciation we explore the distribution of slopes within each catchment.

Hillslope gradients of unglaciated basins tend to be normally distributed about mean and modal slopes that range from ~5-25° (Fig. 4c). In comparison, previously glaciated basins show higher mean and modal slopes >25° with a negative skew towards

low values. Furthermore, we find that these two domains also show clear distributions of slope with elevation. We segmenting each catchment into distinct elevation bins between 50 m contours, and determined the relationship between mean slope angle and mean elevation within the bins (Fig. 5). Dissimilar patterns emerge in how slope varies with elevation within previously-glaciated and non-glaciated catchments. For example, high gradient hillslopes within the non-glaciated basins tend to occur at the upper portions of these basins, well above the mean elevation. However, the steepest hillslopes of glacially sculpted basins

are found at elevations well below the mean (< 1,500 m elevation compared to average elevations of ~1,800 m). This detailed distribution of slope and elevation within glaciated basins is not consistent with the general trend of increasing mean basin slope with mean basin elevation across the study area (Fig. 3). The distribution of slope by elevation within basins (Fig. 5) therefore represents a local signal not reflective of the larger regional trend, and we consider it a fingerprint of past glacial



sculpting, consistent with characteristic glacial and non-glacial slope-elevation curves predicted by Robl et al. (2015). Considering that these previously glaciated basins erode at rates roughly three times faster than average non-glaciated basins, this slope distribution similarly provides a predictive tool for erosion rates (Fig. 5).

## 3.4 Competing controls on Holocene erosion rates

We find compelling evidence of topographic control on erosion; however, other competing hypotheses may explain some of the range of erosion rates found across the region. For example, other climatic controls such as precipitation rates have been invoked to explain fast erosion rates in high peaks of the Alps (Anders et al., 2010). In the Western and Italian Alps, several lines of evidence were used to suggest that post-glacial climates drive the bulk of exhumation and erosion in the region. Multiple studies have suggested that temperature-driven frost cracking processes likely control Holocene erosion rates, based on correlations between elevation and either rock uplift or erosion rates (Delunel et al., 2010; Savi et al., 2015; Vernon et al. 2009). Across our study basins, catchment mean slope and elevation are correlated (Fig. 3), however, elevation poorly correlates with the abundance of steep (>35˚) slopes, notably in the rapidly eroding, previously-glaciated basins. Therefore the elevational proxy for frost cracking does not correspond to topographic indicators of rapid erosion in our study area. Furthermore, we find large differences in erosion rates at basins of the same elevation (Table 2). While frost-cracking may enhance erosion at alpine sites, it does not appear to explain the patterns and variability in erosion rates across our catchments.

Our measured hillslope erosion rates in the Eastern Alps may also be driven by rock uplift and river incision across the region. Previous work has suggested that glaciation during the last glacial maximum (LGM) may drive a Holocene erosional response across the Alps and thereby enhance uplift (Wittmann et al., 2007). Providing a mechanism to engineer this link, Norton et al. (2010a) used observations of correlated river knickpoints and LGM equilibrium line altitudes (ELAs) to suggest that topographic imprint of glacial erosion leads to increased river incision post-glacially, which in turn strengthens the positive feedback between rock uplift and erosion. Could this same mechanism be invoked to explain high erosion rates in our previously glaciated catchments? If catchment erosion were driven by increased river incision, then we should observe higher area-normalized stream gradients in rapidly eroding catchments. Legrain et al. (2015) observed this correlation within the Koralpe region of our study area, but only within small non-glaciated catchments. Therefore, evidence of incision-driven hillslope erosion was found only in the absence of glacial forcings. This finding led these authors to suggest that tectonic uplift in the Eastern Austrian Alps could reasonably explain both 500 m of relief change and a factor-of-three spatial variation in Holocene erosion rates. The scale of uplift (encompassing both the Pannonian basin and entire eastern end of the Alps) suggests deep-seated lithospheric processes (Legrain et al., 2015), and slab detachment (Qorbani et al., 2015) may provide the tectonic mechanism for surface uplift in this Eastern region.





Erosional response to rock uplift may explain local erosional differences within non-glaciated catchments studied here. For example, following Legrain's model, low-elevation catchments in the Styrian basin lie within the incised region below river knickpoints, while higher-elevation catchments in Koralpe with lower erosion rates include significant portions of 'relict terrain'. Importantly, this surface uplift mechanism cannot similarly account for erosional differences between glaciated and

non-glaciated basins. The glaciated basins studied here would fall within the 'relict landscape' region mapped by Legrain et al. (2015) as above river knickpoints, and therefore our high erosion rates do not correlate with area below knickpoints as previously suggested. Furthermore, if uplift drove erosion in these basins, then we would expect to see higher area-normalized stream gradients in more rapidly eroding catchments reflecting the enhanced river incisional response. Figure 6 shows hillslope gradients within each catchment, binned by accumulation area, the upslope and upstream contributing area for all points within

the basin. While mean basin slopes are generally higher in more rapidly eroding glaciated catchments, these higher gradients occur only at the uppermost portions of the catchments – at small upslope accumulation areas less than $\sim 10^2\,\mathrm{m^2}$ that are within the hillslope domain. By comparison, local stream gradients in glaciated and non-glaciated basins are similar at the larger contribution areas (approaching $10^5\,\mathrm{m^2}$ area) that reflect the fluvial domain. The variability in within-basin slope seen only at low contributing areas indicates that the morphological differences within our large catchments studied here are driven by

processes solely within the hillslope domain. The lack of evidence of incision-driven erosion further supports our conclusions that topographic forcings, and not rock uplift, are largely responsible for the patterns in erosion we observe here.

### 3.5 Erosion and Topography across the Alpine Range

While post-glacial topography largely explains the range of erosion rates found at the far end of the Eastern Alps, we note that

these measured erosion rates are still significantly lower than measurements across other regions of the Alps (Fig. 7a). The highest rates measured in our study region are amongst the lowest measured across the Alpine range. Compiling previously reported cosmogenic [10]Be-derived rates across the Alps, we find mean basin slope and Holocene erosion rates are generally weakly correlated ($r^2$=0.26, p<0.001), providing little predictive power for assessing erosion patterns (Fig. 7b) at an orogen scale. This lack of correlation is not surprising at high mean slope angles and rapid erosion rates since erosional processes

become non-linear approaching threshold slope angles (e.g., DiBiase et al., 2010). Poor correlations between most topographic metrics and Alpine erosion rates have been noted before (e.g., Norton et al., 2011; Salcher et al., 2014; Wittmann et al., 2007). Complexities in lithologic variation can partially explain the high scatter in erosion rates at steep gradients (e.g., Norton et al., 2011) since rock strength and fracturing may control slope thresholds. Weaker lithologies often correspond to low hillslope gradients (e.g., Norton et al., 2011) and normalized stream steepness indices (Sternai et al., 2012) in the absence of other

controls. Despite some lithologic influence, orogen-scale controls on Holocene denudation rates have remained relatively elusive.





We might expect Holocene erosion to reflect exhumation and uplift across the range. In the central Alps, some of the observable modern rock uplift has been attributed to a combination of an isostatic response to Holocene erosion (Champagnac et al., 2007; Wittmann et al., 2007) and ice melting (Barletta et al., 2006); however, this latter mechanism has been disputed to drive modern rock uplift (e.g. Persaud and Pfiffner, 2004). Long-term exhumation rates from thermochronometric ages are largely attributed to deep tectonic processes that increased during the Cenozoic (Cederbom et al., 2011), possibly due to slab detachment focused primarily in the west (Baran et al., 2014; Fox et al., 2015; Fox et al., 2014), but also potentially observable in the Eastern Alps (Qorbani et al., 2015). Short-term rates of uplift and erosion and modern topographic metrics appear to poorly or only partially reflect this broad tectonic signal (Koons, 2009; Norton et al., 2011; Vernon et al., 2009), though along-orogen tectonic differences cannot be ruled out in contributing to the variation in erosion rates (Baran et al., 2014).

Climate variability should also be considered in controlling erosion at an orogen scale. Precipitation patterns vary across the range, with highest modern and LGM precipitation occurring in the northern slopes of the Alps and decreasing to the south and east (Florineth and Schlüchter, 1998). Modern precipitation varies from ~400 to >3000 mm/yr across the orogen, and small scale variations in topography have a pronounced effect on local patterns (Isotta et al., 2014). There is reason to believe that modern precipitation gradients should control Holocene erosion and sediment transport by influencing the discharge of sediment out of a basin, controlling landslide thresholds, and by influencing the magnitude of river incision. While similar relationships have been observed across other mountain ranges (e.g., Bookhagen et al., 2005), explicit links between modern precipitation and post-glacial hillslope erosion remain elusive in the Alps (Bennett et al., 2013; Schlunegger and Norton, 2013). However, multiple lines of evidence, including data presented here, suggest that paleoclimate may have a lasting imprint on landscape topography and erosion. Anders et al. (2010) found precipitation is inversely correlated with the elevation of cirque floors in portions of the Swiss Alps, suggesting a climate-driven glacial buzzsaw across the region. Furthermore, glacial erosion during the Pleistocene resulted in notable increases in valley-scale topographic relief (Sternai et al., 2012; Valla et al., 2011). Considering precipitation varies across both small and large scales across the range, this variation is reflected in ice volumes, and the glacially driven topographic legacies persist to modern day, we propose that modern hillslope response to glacial history can partly explain orogen-scale variability in erosion rates. Though focused locally in the Eastern Alps, our new erosion rates and topographic analysis add weight to an increasingly compelling argument that local Holocene denudation rates across the Alps, which often poorly reflect other broader tectonic and climatic controls, are overprinted by the local topographic legacy of glacial sculpting.

## 4 Conclusions

Our study provides repeated evidence that Holocene erosion in the eastern Austrian Alps is driven by glacial legacies that set local topographic forcing and hillslope morphology. Previous work in the region quantified that deep seated tectonic processes



could explain almost a factor-of-three variation in erosion rates in unglaciated terrain (49-137 mm/ka; Legrain et al., 2015). Post-glacial topographic forcings account for an additional doubling over invoked tectonic forcings (resulting in erosion rates averaging 200 mm/ka and up to ~240 mm/ka in previously glaciated basins). Considering that glaciers occupy uplands which are not yet reached by river knickpoints, then this glacial forcing is far in excess of background erosion rates inferred to be

pre-Miocene (49 mm/yr; Legrain et al., 2015). Therefore, despite evidence for young uplift across the eastern extent of the range, glacial processes still dominate the erosion signal, with deeper tectonic forcings likely observable only in the absence of strong local topographic forcings. Our new data suggest post glacial topographic forcing can account for a 4-5x increase over background hillslope erosion rates in the absence of tectonic forcings. Comparison with erosion rates across Alpine range show that these glacially-enhanced rates are still among the lowest measured across the Alpine orogen, and that combined

complexities in tectonic forcings (e.g., Wagner et al., 2010), modern and past climatic forcings (e.g., Anders et al., 2010) and transient erosional response to inherited topographic legacies (this study) must all be considered to understand orogen-scale controls on Alpine Holocene erosion.

## Acknowledgements

This work was funded by the Topo-Alps Project (German National Science Foundation DFG grant BL562/8 to FvB). JLD

acknowledges support from a Montana Institute on Ecosystems award from NSF EPSCoR EPS-1101342. The authors thank Nicolas Legrain for contributions in the field and lab and Hella Wittmann for laboratory support.

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



**Figure 1.** (A) The study area lies in the easternmost region of the European Alps, with sites located across both Austria and Northern Slovenia. (B) Catchment outlines and sampling points for our 26 river sand samples, used to measure millennial scale erosion rates across both previously glaciated and unglaciated catchments. Extent of Last Glacial Maximum (LGM) ice is shown by the shaded area. The boundary between the crystalline bedrock of the Alps and the Miocene sedimentary basin

5 (Styrian Basin) follows roughly the 300 – 500 m elevation contour marked by the yellow color tones.





**Figure 2.** Geographic distribution of erosion rates. (A) Study catchments span several geologic regions marked by distinct massifs and basins. Catchment outlines and data points are color-coded by region. Dashed outlines represent basins that were previously glaciated during the LGM. Extent of LGM ice is shown by the thin grey line, and bolder grey lines mark national boundaries. (B) Erosion rates increase with mean basin slope and can generally be grouped by region (replicate samples shown).

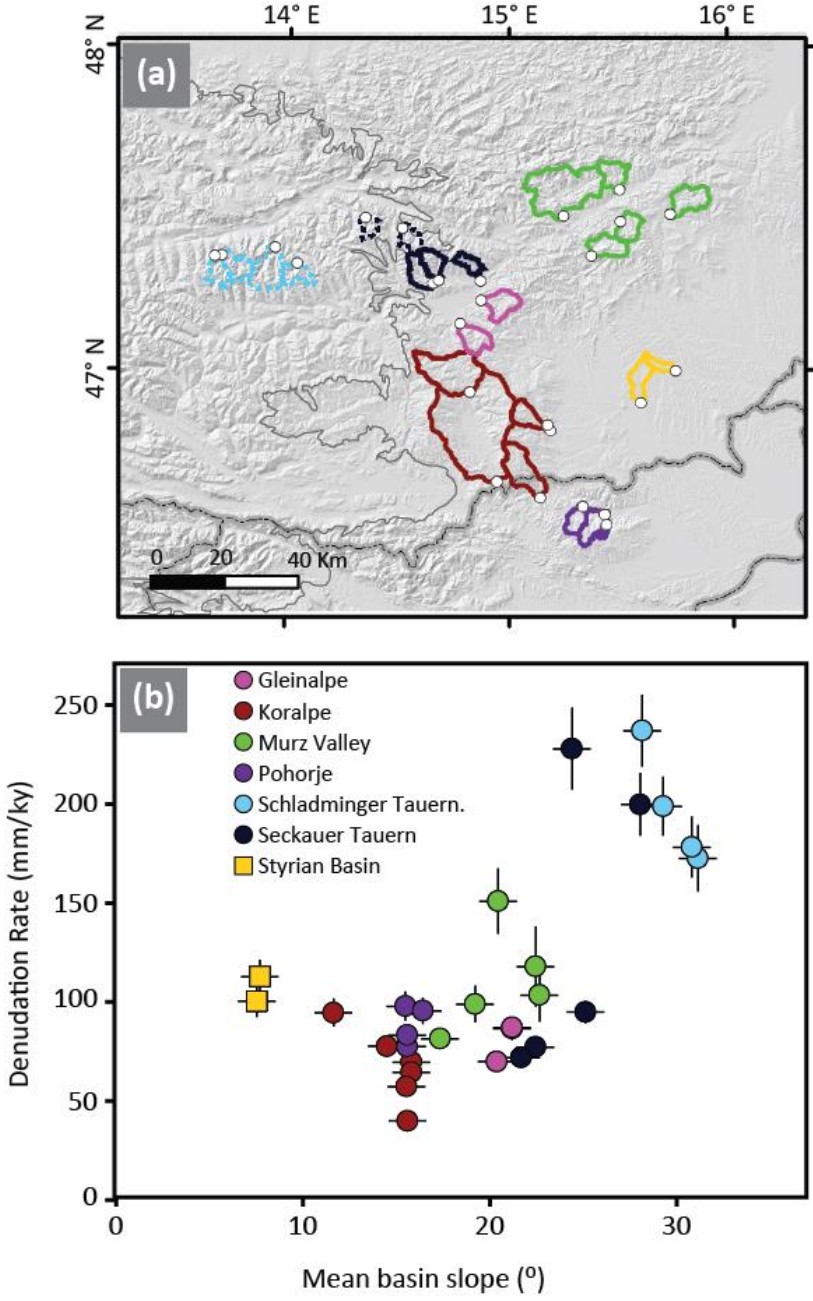



**Figure 3.** Study catchments show increasing mean slope with mean elevation (black circles). The percent of slopes >35° within these basins increase non-linearly with elevation (blue squares), such that catchments at high elevations >1500 m show strong variation in the distribution of steep, threshold-style hillslopes. Small unglaciated basins studied by Legrain et al. (2015) in this same region show little systematic variation in slope with elevation (small open circles). Instead of reflecting the broader

5  regional signal, mean slopes of these smaller basins are likely controlled by their position with respect to river knickpoints and proportion of the catchment that is actively incising.

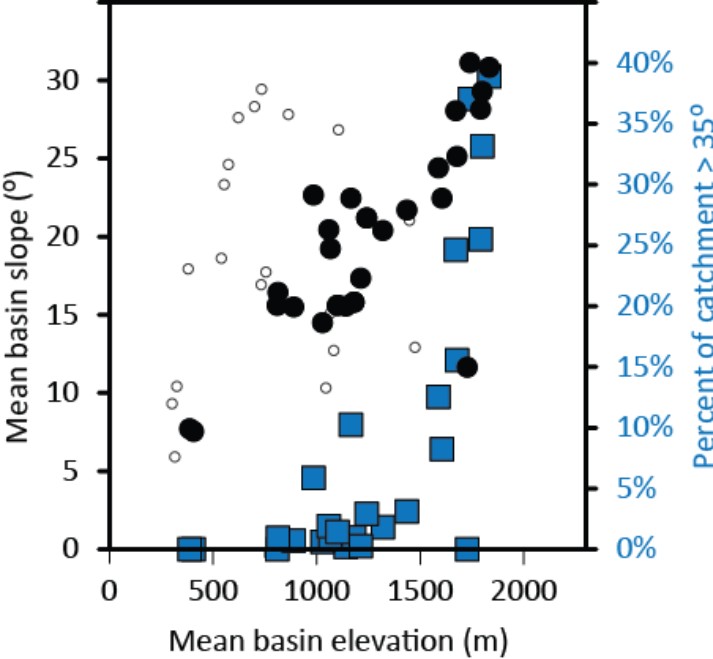





**Figure 4.** (A) Map of sampled basins across the eastern Alps, color-coded for erosion rates. (B) Cumulative slope distributions color-coded by sample-region (same as Figure 2) show catchment morphology follows geographic groupings, with low-slope endmembers in the Styrian basin and high-slope endmembers represented by previously-glaciated basins of the Schladminger and Seckauer Tauern (dotted lines). Slope distributions across these basins also complement measured erosion rates. (C,D) Frequency and cumulative distributions of basin slope show that rapidly eroding, previously-glaciated basins tend to have higher mean and modal slopes than more slowly eroding basins. Colors in panels C and D correspond to the scale for basin erosion shown in panel A.

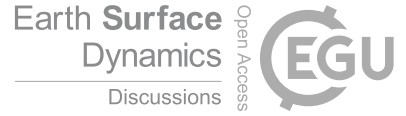

**Figure 5.** Variation of slope and elevation within sample catchments. Elevation is binned between 50 m contours, and catchment slopes are averaged within these bins. Hillslope angles are distinctly distributed with basin elevation between slow and rapidly eroding catchments; with highest slope angles found at middle elevations of previously glaciated basins and highest elevations of non-glaciated basins. Symbol colors correspond to the scale for basin erosion shown in Fig. 4a.

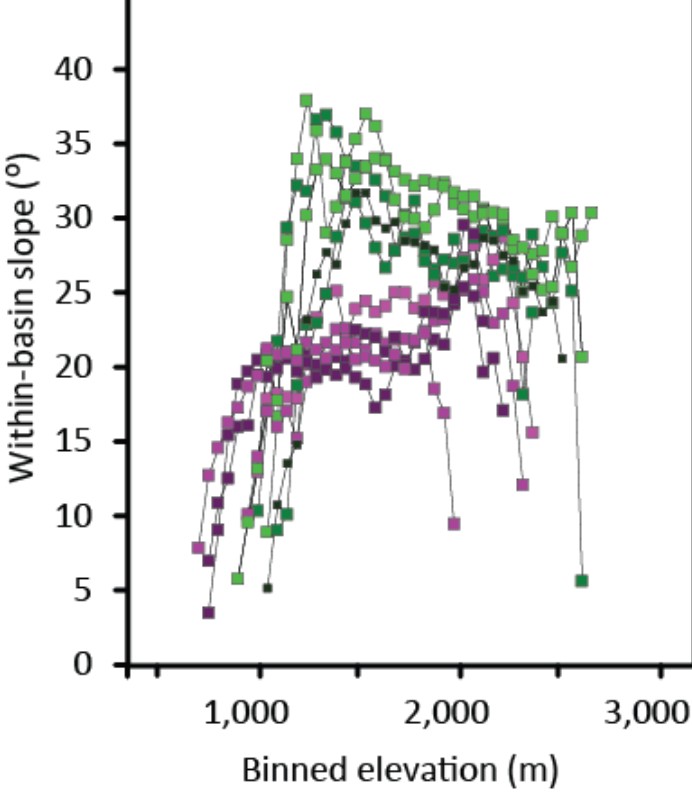



**Figure 6.** Slope-area plots for catchments across the study area. Accumulation area is calculated on a per pixel basis and represents the upslope contributing area (drainage area). Slopes within each catchment are binned by increments of 0.2 $\log_{10}$ accumulation areas to show downslope and downstream changes in mean basin gradient. Binned values are color coded by erosion rate, and correspond to the basin erosion scale provided in Fig. 4a. Data points at large accumulation areas ($>10^3$ m$^2$) reflect local stream steepness, and plot within a similar range of values despite disparate erosion rates. However, data points at small accumulation areas ($<10^2$ m$^2$) represent upslope hillslope gradients and have distinct steepnesses based on the erosion rates of the basin and the glacial history. These data largely reflect disparate hillslope steepnesses between glaciated (rapidly eroding; green) and unglaciated (more slowly eroding; pink) catchments.

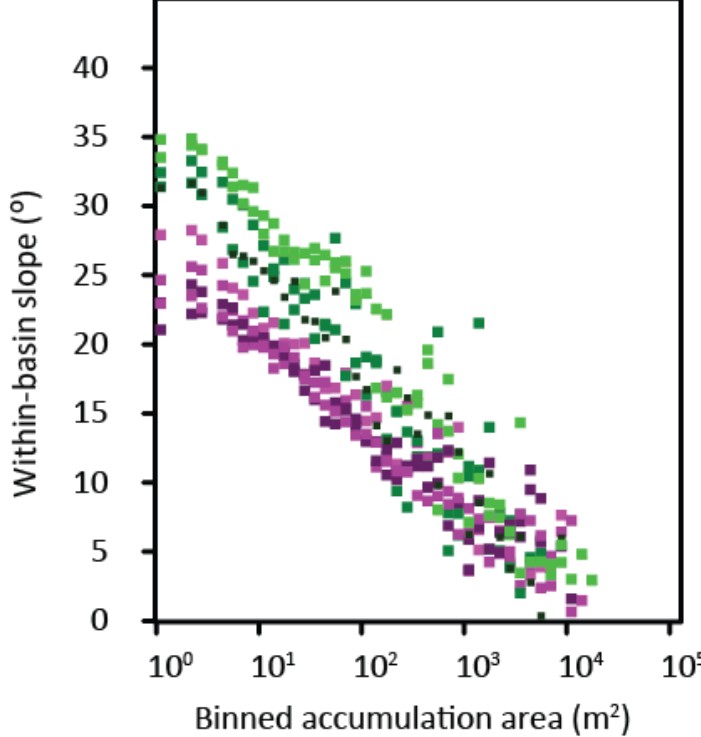





**Figure 7.** (A) Published erosion rates across the European Alps range from ~40 - 2,100 mm/ka, recorded by over 100 cosmogenic samples from 7 studies that report both mean catchment slope and erosion rates. Symbol size reflects the published erosion rate, and symbol color reflects past glacial history (red = previously unglaciated; blue = previously glaciated). (B) Across the range, these rates vary only weakly with mean basin slope. Rates from Legrain et al., 2015 were measured in small basins with areas <1 km², and are only shown in panel B.





**Table 1.** Data from cosmogenic nuclide analyses

| Sample | Size Fraction (µm) | Long. (°E) | Lat. (°N) | Sample elev. (m) | Qtz. weight (g) | [10]Be conc.* (x10[4] at/g qtz) | Snow shielding factor [#] | Topographic shielding factor [#] | Production rate † (at / g qtz/y) | Denudation rate (mm/ky) |
|---|---|---|---|---|---|---|---|---|---|---|
| Bistrica | 250-500 | 15.139 | 46.613 | 382 | 30.2 | 10.87 ± 0.89 | 0.93 | 0.99 | 9.58 | 69.2 ± 6.4 |
| Bistrica | 500-800 | 15.139 | 46.613 | 382 | 30.2 | 11.74 ± 0.87 | 0.93 | 0.99 | 9.58 | 64.0 ± 5.6 |
| Feistrichbach | 250-500 | 14.778 | 47.151 | 717 | 31.2 | 12.79 ± 0.60 | 0.92 | 0.98 | 11.63 | 69.3 ± 4.2 |
| Gleinbach | 250-800 | 14.872 | 47.222 | 656 | 33.2 | 9.41 ± 0.56 | 0.92 | 0.98 | 10.46 | 86.3 ± 6.1 |
| Gleinbach | 500-800 | 14.872 | 47.222 | 656 | 35.1 | 9.35 ± 0.45 | 0.92 | 0.98 | 10.46 | 86.8 ± 5.0 |
| Ingering | 250-500 | 14.680 | 47.285 | 967 | 40.3 | 11.39 ± 0.56 | 0.89 | 0.96 | 14.50 | 94.3 ± 5.8 |
| Kleinsolk | 250-500 | 13.938 | 47.382 | 893 | 39.9 | 6.58 ± 0.57 | 0.88 | 0.93 | 15.53 | 172.5 ± 16.2 |
| Krug | 250-500 | 14.657 | 47.275 | 938 | 40.2 | 13.81 ± 0.78 | 0.90 | 0.97 | 14.24 | 76.5 ± 5.4 |
| Lassnitz1 | 250-500 | 15.184 | 46.820 | 520 | 34.6 | 15.49 ± 0.62 | 0.96 | 0.99 | 7.48 | 39.2 ± 2.1 |
| Lassnitz3 | 250-500 | 15.173 | 46.838 | 543 | 31.7 | 13.70 ± 0.60 | 0.93 | 0.99 | 9.95 | 56.7 ± 3.4 |
| Lavant1 | 250-500 | 14.824 | 46.939 | 658 | 31.7 | 8.95 ± 0.58 | 0.89 | 0.99 | 11.02 | 94.3 ± 7.0 |
| Lavant2 | 250-500 | 14.945 | 46.663 | 370 | 31.5 | 9.95 ± 0.58 | 0.94 | 0.99 | 9.81 | 77.4 ± 5.3 |
| Mooskogel | 250-500 | 15.372 | 47.359 | 490 | 39.0 | 7.04 ± 0.83 | 0.95 | 0.97 | 9.19 | 103.7 ± 13.4 |
| Obertal | 250-500 | 13.665 | 47.354 | 967 | 40.1 | 5.69 ± 0.38 | 0.88 | 0.94 | 15.52 | 199.0 ± 15.0 |
| Pickelbach | 250-500 | 15.750 | 47.003 | 326 | 30.7 | 4.91 ± 0.38 | 0.99 | 1.00 | 5.65 | 101.6 ± 8.3 |
| Pohorju1 | 250-500 | 15.434 | 46.529 | 543 | 32.7 | 9.53 ± 0.78 | 0.94 | 0.99 | 9.29 | 77.2 ± 7.1 |
| Pohorju1 | 500-800 | 15.434 | 46.529 | 543 | 31.5 | 8.86 ± 0.47 | 0.94 | 0.99 | 9.29 | 83.1 ± 5.1 |
| Pohorju2 | 250-500 | 15.425 | 46.560 | 313 | 30.6 | 7.22 ± 0.50 | 0.95 | 0.99 | 8.85 | 98.1 ± 7.7 |
| Ratten | 250-500 | 15.732 | 47.487 | 752 | 30.3 | 10.10 ± 0.48 | 0.93 | 0.99 | 10.58 | 81.1 ± 4.8 |
| Rottenmann | 250-500 | 14.347 | 47.477 | 1077 | 35.5 | 5.51 ± 0.39 | 0.89 | 0.95 | 15.04 | 200.1 ± 15.9 |
| SaintNico | 250-500 | 14.038 | 47.335 | 1053 | 40.0 | 4.88 ± 0.33 | 0.88 | 0.95 | 15.93 | 237.5 ± 17.8 |
| Seckaur | 250-500 | 14.871 | 47.283 | 724 | 32.4 | 12.53 ± 0.70 | 0.91 | 0.98 | 11.77 | 71.3 ± 4.8 |
| Stanz | 250-500 | 15.506 | 47.465 | 650 | 29.5 | 7.71 ± 0.65 | 0.94 | 0.98 | 9.71 | 99.0 ± 9.3 |
| Stiefing | 250-500 | 15.592 | 46.905 | 304 | 31.2 | 4.35 ± 0.31 | 0.99 | 1.00 | 5.62 | 114.3 ± 8.5 |
| Thorl | 250-500 | 15.245 | 47.482 | 583 | 40.3 | 6.49 ± 0.97 | 0.91 | 0.97 | 9.81 | 118.2 ± 19.6 |
| Triebental | 250-500 | 14.516 | 47.444 | 1063 | 32.2 | 4.50 ± 0.38 | 0.90 | 0.96 | 13.95 | 228.7 ± 21.8 |
| Untertal | 250-500 | 13.698 | 47.357 | 999 | 40.5 | 6.53 ± 0.50 | 0.88 | 0.93 | 16.00 | 178.1 ± 15.4 |
| Veitsch | 250-500 | 15.502 | 47.564 | 672 | 34.2 | 4.96 ± 0.51 | 0.94 | 0.98 | 9.55 | 151.8 ± 16.5 |
| Velka | 250-500 | 15.328 | 46.585 | 372 | 34.3 | 6.91 ± 0.45 | 0.96 | 0.99 | 8.13 | 95.8 ± 6.8 |

* [10]Be concentrations measured at ETH-Zürich in June 2010 and 2011. Results normalized to Nishiizumi et al. (2007) 2007KNSTD standard, corrected for average of six chemical processing blanks ([10]Be/[9]Be = 2.72 ± 2.21 × $10^{-15}$; µ ± s.d.).

[#] Snow shielding calculated from annual Swiss snow data (Auer, 2003). Topographic shielding calculated from 10 m DEMs.

† Per-pixel production rates calculated for quartz-bearing lithologies following scaling laws of Dunai (2000), Schaller et al. (2002), and Braucher et al. (2003) for nucleonic and muonic interactions. Based on compilation of high-latitude, sea-level production rates of 4.0 atoms/g$_{quartz}$/yr for high-energy neutrons (Phillips et al., 2016), and assuming negative and fast muons compose 1.2% and 0.65% of total production rates respectively (Braucher et al., 2003). Mean catchment production rates include both topographic and snow-shielding correction factors.



**Table 2.** Catchment denudation rates and morphometrics

| Region and Sample | | Glacial History * | Catchment Area (km2) | 10m slope (°) | 80m slope (°) | Mean Elevation (m) | Denudation Rate (mm/ky) |
|---|---|---|---|---|---|---|---|
| Gleinalpe | | | | | | | |
| | Feistrichbach | U | 74.8 | 20.4 | 19.5 | 1322 | 69.3 ± 4.2 |
| | Gleinbach | U | 80.9 | 21.2 | 20.4 | 1243 | 86.3 ± 6.1 |
| Koralpe | | | | | | | |
| | Bistrica | U | 141.2 | 15.8 | 15.1 | 1181 | 69.2 ± 6.4 |
| | Lassnitz1 | U | 2.9 | 15.6 | 15.5 | 810 | 39.2 ± 2.1 |
| | Lassnitz3 | U | 66.3 | 15.5 | 14.8 | 1141 | 56.7 ± 3.4 |
| | Lavant1 | U | 234.5 | 11.6 | 11.4 | 1728 | 94.3 ± 7.0 |
| | Lavant2 | U | 952.3 | 14.5 | 14.2 | 1029 | 77.4 ± 5.3 |
| | Mooskogel | U | 79.2 | 22.7 | 22.4 | 985 | 103.7 ± 13.4 |
| Murz Valley | | | | | | | |
| | Ratten | U | 96.5 | 17.3 | 16.8 | 1213 | 81.1 ± 4.8 |
| | Stanz | U | 62.3 | 19.2 | 19.0 | 1068 | 99.0 ± 9.3 |
| | Thorl | U | 321.1 | 22.5 | 21.9 | 1166 | 118.2 ± 19.6 |
| | # Thorl-qtz | U | (163.7)* | - | 10.9 | 1141 | 118.2 ± 19.6 |
| | Veitsch | U | 73.0 | 20.4 | 20.1 | 1060 | 151.8 ± 16.5 |
| | # Veitsch-qtz | U | (63.3)* | - | 19.0 | 814 | 151.8 ± 16.5 |
| Pohorje | | | | | | | |
| | Pohorju1 | U | 10.9 | - | 14.8 | 1102 | 77.2 ± 7.1 |
| | Pohorju2 | U | 78.3 | - | 14.6 | 890 | 98.1 ± 7.7 |
| | Velka | U | 52.5 | - | 16.0 | 814 | 95.8 ± 6.8 |
| Schladminger Tauern | | | | | | | |
| | Kleinsolk | G | 117.8 | 31.1 | 30.6 | 1742 | 172.5 ± 16.2 |
| | Obertal | G | 51.8 | 29.3 | 28.9 | 1800 | 199.0 ± 15.0 |
| | SaintNico | G | 60.3 | 28.1 | 27.4 | 1792 | 237.5 ± 17.8 |
| | Untertal | G | 67.8 | 30.8 | 30.3 | 1834 | 178.1 ± 15.4 |
| Seckauer Tauern | | | | | | | |
| | Ingering | U | 58.6 | 25.1 | 24.4 | 1679 | 94.3 ± 5.8 |
| | Krug | U | 67.0 | 22.4 | 21.9 | 1606 | 76.5 ± 5.4 |
| | Rottenmann | G | 32.2 | 28.0 | 27.2 | 1672 | 200.1 ± 15.9 |
| | Seckaur | U | 37.7 | 21.7 | 21.2 | 1437 | 71.3 ± 4.8 |
| | Triebental | G | 54.8 | 24.4 | 23.6 | 1590 | 228.7 ± 21.8 |
| Styrian Basin | | | | | | | |
| | Pickel | U | 27.7 | 7.5 | 7.0 | 406 | 101.6 ± 8.3 |
| | Stiefing | U | 66.6 | 7.7 | 7.2 | 388 | 114.3 ± 8.5 |

\* Catchments defined as previously glaciated (G) or unglaciated (U) in the Pleistocene.

\# Thorl and Veitsch catchments contained significant non-quartz bearing lithologies. Topographic metrics for quartz bearing regions alone provided separately.