# Peer review of "Glaciation's topographic control on Holocene erosion at the eastern edge of the Alps"

_Earth Surface Dynamics, 2016_

## Short Comment (SC1) · 27 Jun 2016

Marco G. Jorge

mjorge@sfu.ca

I find the manuscript very interesting and look forward to its final version. Below, please find some comments and proposed edits that may help improving the manuscript.

—- Introduction —-

Page 2, Sentence L3-5. Suggest rephrasing. Second clause does not preclude first clause.

L7. Perhaps, the transition to glacial buzzsaw concept is not properly backgrounded by previous sentence.

L13. Suggest rephrasing for flow with previous sentence. E.g., first say that glacial landscapes can be preserved . . ., influencing. . .

[Figure]

L15. Widely glaciated mountain belts?

L16. References missing.

L18. Rephrase: not all glaciers are. . .

L25. Complete "Climate's influence on . . .

—- Section 2.3, Digital Terrain Analysis —-

Page 4, L31. Suggest mentioning that SRTM is 1 arc-second, instead of 80 m grid (dependence of cell size on latitude).

Page 5, L4-5. This remark, specifically the last sentence, is too precise, or off what would be reasonable to discuss based solely on the differences in the topographic metrics. Basin slope gradient average is not indicative of local scale morphometry independently of DEM cell size and an average gives no information about spatial variation. Either remove or rephrase and extend this discussion by adding more information (references?).

(Table 2: refer that slope gradient values are averages)

Which parameters were used for stream and catchment delineation? Was a flow accumulation threshold used? I presume from Fig 1 that delineation was based on the location of sampling sites (basin outlet). I believe it is important to justify location of sampling sites as well as its influence on basin delineation and morphometry.

—- Section 3, results and discussion —-

The message would be clearer and the manuscript a better read if the discussion was separated from the results; there would be less back and forth. The discussion would benefit from the inclusion of further morphometrics (e.g., of elevation dispersion). In instances, the conclusions within the discussion overshoot what would be reasonable to conclude from the presented data (see below). I think that it is important to include an evaluation of lithology as a conditioning factor of the observed differences in erosion

rates (even if it is null).

Page 5,

L11. which lowland basin catchments? In the Styrian Basin?

L22. Rephrase. "these data" refers to both mean elevation and slope gradient but this sentence and following sentences address slope only.

L23. Whereby –> where

L23, 24. The described relationship between mean slope and erosion rates does not imply non-linear relationship. Perhaps reword results.

Page 6,

L11-13 (paragraph's last sentence) Recommend rephrasing. Remove first clause and reword last clause.

L16-21. This is too simplistic. For example, note that glaciated catchments generally are higher in elevation and non-glaciated catchments vary widely in mean elevation (Table 2) (differences in potential energy).

L25. segmenting –> segmented

L23-36. Too simplistic and somewhat confusing. Differences in average elevation between basins and elevation-slope relationships within basins are different things. Why should the relative location of the steepest slopes be positively related to basin average slope gradient? Justification for interpreting that to be signal of past glacial sculpting is insufficient.

Page 7,

L11. 'However' should be preceded by semi-colon.

L14-16. Is the abundance of slopes >35° in gradient a good proxy for frost cracking? Address it directly.

L24, 25 (Sentence). Explain; and what are area-normalized stream gradients? (area of what?)

L26, 27 (Sentence). Add reference.

Last paragraph. What is the authors' take on this discussion?

Page 8,

L5-7. It was referred before that Legrain et al., 2015 looked at non-glaciated basins. Does "previously suggested" refer to Legrain et al., 2015?

—- Conclusions —-

Page 9,

L1. "Repeated" meaning supporting previous studies? –> Add references

Page 10,

L2, 3. Not clear where these values are from; add references?

---

## Referee Comment (RC1) · P. Valla (Referee) · 29 Jun 2016

Dear Authors, dear Editors,

Please find below my review concerning the manuscript by Dixon and co-authors entitled "Glaciation's topographic control on Holocene erosion at the eastern edge of the Alps" (Paper # ESurf-2016-29).

This manuscript investigates the spatial variability and potential topographic controls on Holocene erosion rates in the eastern Alps. The authors present original catchment-wide erosion rates (26 new basins) from cosmogenic 10Be at the eastern edge of the Alps (Austria and Slovenia). They compare their erosion results with detailed topographic metrics to discuss any potential climatic, tectonic and/or topographic control on the spatial distribution they observe for catchment-wide erosion rates. Finally, they

combine their new results with existing 10Be-derived catchment-wide erosion rates all over the European Alps to discuss the different forcings on Holocene erosion at the orogen-scale.

This is a very interesting, well-written and presented manuscript. The authors have really well introduced their work with a comprehensive review of the literature and presentation of the open questions concerning the late evolution of the European Alps. They propose new detrital 10Be-derived erosion rates that nicely complement previous observations in the Eastern Alps and have performed a detailed topographic analysis to discuss their dataset and the spatial variability of catchment-wide erosion rates over the entire European Alps. As such this study represents in my opinion a valuable contribution and will have potential of great interest for the community. I have some questions and suggestions, mainly to better clarify some of the results and their interpretation, and have outlined them in a set of general and specific comments below.

General comments: 1 - Structure: the present manuscript is overall well-structured, with clear "Introduction" and "Approach" sections. However, the merged section 3 "Results and Discussion" is sometimes quite difficult to follow between the different subsections. I would suggest the authors to separate the results from the discussion which may be easier to follow for the readers.

2 - 10Be-derived basin erosion rates: the new results for 26 basins presented by the authors are really interesting and nicely complement previous investigations of Holocene erosion rates across the European Alps. However, I would recommend the authors to provide more discussion about the possible bias in calculated erosion rates from local complexities and the resulting implications for their story. I have listed some detailed questions below, especially concerning the integration times, snow-cover correction, the sediment grainsize, the approach for floodplains and non-quartz bearing areas. . . Moreover, some discussion about the (previously) glaciated catchments would be helpful for ESurf readers concerning the re-mobilization of morainic/glaciogenic material, or the glacial perturbation on 10Be concentrations which might be non-negligible for

**ESurfD**
slowly-eroding terrains (e.g. Glotzbach et al., 2014 Terra Nova). Finally, the compilation of existing 10Be-derived basin erosion rates from the literature is very interesting and nicely put the new dataset into a broader context. I am wondering if 10Be-derived basin erosion rates have been directly taken from published papers, or if the authors have recalculated the erosion rates with a uniform/updated production rate? If not, maybe discuss also this point and potential implications.

3 - Topographic metrics: the authors provide a detailed topographic analysis of the studied catchments with a special attention on mean basin slopes and slope distributions within the basins. I would suggest to also report other metrics, such as hypsometry or local relief to strengthen their message (see also specific comments below). Finally, the authors discussed in some sections the role of fluvial incision and river response to potential perturbations, but this would be better illustrated with some river profiles shown as a new figure. This will clearly help readers to evaluate the degree of disequilibrium in river profiles, and potential differences between regions/massifs or glaciated vs. non-glaciated catchments.

Specific comments, by line number: - Page 1, lines 26 and 27: "poor topographic indicators of controls" (l.26) contradicts with the end of the next sentence "its topographic legacy" (l.27). Maybe rephrase the sentence on line 26 (which also appears in contradiction with the manuscript's title).

- Page 2, lines 14-15: "suggest that glacial forcings are the dominant control on landscape evolution in mountain belts". This sentence reads vague and quite general, maybe rephrase to precise in which mountain belts (mid- and high-latitudes?) and over which timescales (Plio-Quaternary?).

- Page 2, line 26: "have been invoked as principle drivers of erosion and uplift". Maybe cite also there Fox et al. 2015 (Geology).

- Page 2, line 28: "post-glacial erosion may explain rates of uplift in the region via isostasy". Please clarify the timescales over which this correlation is valid, i.e.

**ESurfD**
Holocene for erosion rates from Wittmann et al. (2007) and historical/modern for uplift rates (leveling data from Schlatter et al., 2005).

- Page 2, line 29: "youthful tectonic features such as river knickpoints". Please rephrase, river knickpoints can have several origins and as rightly stated after ("correlated with previous glacial cover...") these are not tectonic markers.

- Page 3, line 8: "erosion rates". Please specify the timescale "post-LGM/Holocene" for these erosion rates.

- Page 3, lines 13-14: Please indicate the massifs, basins and other important locations on Figure 1 for clarity.

- Page 3, line 17: "that has uplifted some 300 m above sea level in the last 7 Ma". Please add a reference here if possible.

- Page 3, line 18: "Massifs" corrected by "massifs".

- Page 3, line 27: "This timing coincides with inversion and uplift of the Styrian and...". Please clarify which timing is considered here. Is it 4 Ma (l.27), 7 Ma (l.17) or 10 Ma (l.16)?

- Page 4, line 9: "250-500 um size fraction". In Table 1, some sample sizes are higher (500-800 um). Are they replicates? Please clarify, and maybe also discuss potential implications when comparing the resulting erosion rates for different grainsizes.

- Page 4, line 25: "elevation-snow depth relationships previously determined in the Swiss Alps". I am not sure to what extent the data from Auer (2003) in the Swiss Alps can be extrapolated to the Austrian/Slovenian Alps (this extrapolation might depends on the local precipitations and moisture patterns). How comparable are the two climatic settings? Please discuss this extrapolation and the potential implications for calculated erosion rates.

- Page 4, line 26: "we set production rates equal to zero in parts of drainage basins with

carbonate terrains". These areas should be excluded from the calculations (since they do not contribute quartz) and not set to zero-production, otherwise this would introduce a bias in the catchment-wide erosion rates. Please correct or clarify. Also, how did the authors consider low-gradient areas such as floodplains (figure 1) in their calculations? Are they included in the integrated 10Be erosion rate calculations? Please clarify.

- Page 5, line 14. The (previously) glacial setting for these catchments may imply potential biases in the 10Be concentrations and thus in the calculated erosion rates, with input from morainic/glaciogenic material (e.g. Delunel et al., 2014 ESPL) or the impact of former glaciation (Glotzbach et al., 2014 Terra Nova). I would suggest the authors to discuss these points further and to what extent they may perturb the inferred catchment-wide erosion rates.

- Page 5, line 21: "Measured erosion rates also generally increase with increasing mean basin elevation". Can the authors provide a figure for this correlation?

- Page 5, line 22: "Measured erosion rates also generally increase with [. . .] slope". Looking at Figure 2b, we can also see two clusters: 1) non-glaciated basins with no correlation between slope and erosion, and 2) (previously) glaciated basins where there seem to be an inverse correlation between slope and erosion. Please consider discussing this potential alternative observation.

- Page 5, line 28: "We note that both erosion rates and catchment mean slope correlate with the proportion of the catchment that exceeds 35° (Fig. 2b)". This is not shown on figure 2b, or maybe I missed something. Please correct or clarify.

- Page 6, lines 9-14: "therefore likely reflect this erosional response to river incision and tectonic processes across the range". In the previous section (3.1, lines 11-12), the spatial variability in erosion rates was suggested to reflect lithological variations. Please clarify. - Page 6, line 25: "We segmenting". Please correct.

- Page 7, line 1: "consistent with characteristic glacial and non-glacial slope-elevation

curves predicted by Robl et al. (2015)." This specific slope distribution with elevation for glaciated terrains has already been observed in other places (e.g. van der Beek and Bourbon, 2008 Geomorphology). Please cite some references.

- Page 7, line 7: "other climatic controls such as precipitation rates". How variable are the mean precipitation rates between the different studied catchments? Please discuss this point.

- Page 7, line 11: "correlations between elevation and either rock uplift or erosion rates". Unclear, please clarify. I think this would rather be "correlations between elevation or rock uplift and erosion rates". Also, correlations from Vernon et al. 2009 are based on thermochronology and thus imply much longer timescales than Holocene, and they do not consider frost-cracking. Please correct.

- Page 7, line 13: "elevation poorly correlates with the abundance of steep (>35Ì́Ł) slopes". On Figure 3, there seem to be some non-linear correlation between elevation and fraction of the basin >35°. Please rephrase or discuss further this point.

- Page 7, line 16: "While frost-cracking may enhance erosion at alpine sites, it does not appear to explain the patterns and variability in erosion rates across our catchments". Maybe frost-cracking is occurring (or had occurred in the Lateglacial period) for previously-glaciated catchments, but for non-glaciated catchments the mean elevations are too low to consider this effect. Would it be possible?

- Page 7, lines 24-25: "If catchment erosion were driven by increased river incision, then we should observe higher area-normalized stream gradients in rapidly eroding catchments". What are "area-normalized stream gradients"? Also, did the authors study the river profiles to identify such perturbations. I would suggest the authors to show some river profiles and/or hypsometric curves for the studied catchments to illustrate the discussion about river incision (see also my general comment). Same question for the sentence on Page 8, line 7-8. This is difficult to see the fluvial domain on figure 6, so I would encourage the authors to show some figures focused on the

fluvial part of catchments (river profiles, slope-area diagrams).

- Page 8, lines 3-6. How about lithological variations (and thus erosional resistance) as a potential control on the spatial variability in erosion rates?

- Page 8, line 15: "processes solely within the hillslope domain". Did the authors also look at local relief as a potential topographic metrics for erosion rates?

- Page 8, line 22: "Compiling previously reported cosmogenic 10Be-derived rates across the Alps". Did the authors report here the original erosion rates or did they recalculate erosion rates with uniform/updated production rate? Please clarify and discuss potential implications for comparing erosion rates across the Alps.

- Page 9, line 1: "we might expect Holocene erosion to reflect exhumation and uplift. . .". How can erosion reflect exhumation and uplift? This appears unclear, please rephrase.

- Page 9, lines 4-7: "Long-term exhumation rates from thermochronometric ages are largely attributed to deep tectonic processes that increased during the Cenozoic". Please rephrase, long-term exhumation rates are also driven by Plio-Pleistocene changes in erosion following climatic forcing as well as drainage modifications, not only tectonics.

- Page 9, lines 12-13: "with highest modern and LGM precipitation occurring in the northern slopes of the Alps and decreasing to the south and east". Moisture patterns have changed between the LGM and modern (Florineth and Schluchter, 1998) so I am not sure that precipitation maxima have always been on the northern Alpine slopes. Please clarify.

- Page 10, line 3: Please correct "mm/ka" by "mm/ky" for consistency.

- Page 10, line 5: "49 mm/yr" would rather be "49 mm/ky". Please correct.

Tables and Figures: - Figure 1: I would suggest to add main massifs, basins and maybe river names on Figure 1b to help the readers following section 2.1 and to link
with subsequent figures.

- Figure 2: Please indicate replicates on Figure 2b with a star or different symbol. Also, I would suggest to also add the data from Legrain et al. (2015) on this figure, that may be helpful to compare them already at this stage, no (as they appear on figure 3)?

- Figure 3: Maybe use different symbols (or open/filled) to differentiate between glaciated/non-glaciated basins?

- Figure 4: Why is the "basin erosion" legend reversed in panel a? Panels c and d are nice and informative, I am wondering if similar panels with elevations would be informative? On panel b, what are the criteria for "partially glaciated" and how does it relate to figure 2 with glaciated/non-glaciated? Please clarify.

- Figure 5: Would it be possible to use different symbols for glaciated/non-glaciated catchments?

- Figure 6: This figure is difficult to read at present. I think that the slope distribution for the hillslope domain is already illustrated by panels c and d of figure 3, so I would recommend to show here only the fluvial domain (>103 m2) to better highlight any differences between the different rivers.

- Figure 7: I would be curious to see if there is any correlation between mean basin elevation and erosion rate across the Alps. Did the authors look at this or can add the corresponding figure if informative?

- Table 1: I would suggest to also indicate the "integration time" for the reported denudation rates. In the footnote, please correct "negative and fast muons" by "slow and fast muons".

- Table 2: In the footnote, please replace "Pleistocene" by "LGM". What is the maximum ice coverage during the Pleistocene (do we have evidence ofr more extended glaciations before the LGM in this part of the Alps)?

I hope these comments and suggestions may be useful for revising the manuscript, and I look forward to seeing it published.

Sincerely,

Pierre Valla

Lausanne, 29 June 2016
* * *
**ESurfD**

---

## Referee Comment (RC2) · P.A. van der Beek (Referee) · 7 Jul 2016

Dixon et al. provide new detrital cosmogenic 10Be data to constrain erosion rates of nearly 30 catchments in the easternmost Alps (Austria and Slovenia). While earlier studies in this area have argued for a tectonic control on erosion rates, with catchments influenced by recent uplift recording higher rates than catchments to which this recent phase has not (yet) been communicated, the extended dataset presented here shows that the main controlling parameters on erosion rates are basin relief and mean slope, which the authors argue to be influenced by glacial preconditioning.

This study provides interesting new data that significantly tone down previous interpretations, and provides an integrated view of Holocene erosion rates in the Alps. It is therefore timely and definitely suitable for publication in Earth Surface Dynamics. While

my overall evaluation of this manuscript is thus positive, I recommend it be returned to the authors for moderate revisions before final acceptance. These pertain to some apparent misconceptions or imprecisions in the writing, as well as the intriguing slope-area relationships that may merit some more discussion. As most of my comments are rather specific, I will list them tied to page and line numbers below:

Page 1, line 13: the Hergarten et al. model is based on a fundamental misconception: it mistakes a glacial imprint on topography for a transient tectonic signal. It would be preferable if this fundamentally flawed study were not perpetuated in the literature any more than it needs to; I would thus suggest the authors to refrain from citing it, particularly in the abstract.

Page 1, line 23: Although Legrain et al. do invoke "deep lithospheric processes", it is not sure these are required for the easternmost Alps. In contrast to the west, convergence is still active in the East (e.g. Serpelloni et al., Geophys. J. Int., 2005) and the inversion of the Pannonian basin can be linked to a change in crustal stress fields from extension to compression (itself possibly linked to a deep lithospheric cause, however).

Page 2, line 7. The process of valley deepening and widening described above is not, in fact, the cause of the "glacial buzzsaw". The generation of widespread low-relief surfaces at elevations around the average Quaternary ELA (the topographic fingerprint of the "buzzsaw") is rather linked to efficient cirque retreat, possibly aided by periglacial (frost-cracking) processes. See Mitchell and Montgomery (Quat. Res., 2006) and Egholm et al. (ESurf., 2015) for discussions of these processes.

Page 2, line 9: Isostatic rebound will cause rock uplift but will not in itself increase relief. Relief increase is due to the fact that glacial erosion is strongly non-uniform or "selective", deepening valleys while having limited effects on higher parts of the landscape.

Page 2, line 14: Norton et al. (Geology, 2010) would be a good complementary (or

alternative) reference here.

Page 2, lines 26-27: This presentation of the findings of Wittmann et al. (2007) is slightly misleading. In fact, their regression of denudation rates versus rock-uplift rates gave a slope of 1.0±0.25, i.e. erosion rates could be either higher or lower than rock-uplift rates, and these authors include a lengthy discussion of the potential implications of this finding. Champagnac et al. (2009) did subsequently argue, based on a subset of this data, that rock-uplift rates were lower than denudation rates, but even their analysis is not equivocal on this point.

Page 3, line 1: again, why do you need to invoke deep lithospheric processes in a region where convergence is still ongoing?

Page 3, lines 12-14: there are several regional names here (Styrian (Alps?), Levanttal Alps, Gleinalpe, Koralpe, Schladmig Tauern, Seckauer Tauern, Pohorje) that are not know to a non-Austrian readership. They should be indicated on the map of Fig. 1.

Page 3, lines 16-18: a geological map might make this description of the regional geology easier to follow.

Page 3, lines 27-28: "but appears conspicuously unrelated" to what? This is unclear . . .

Page 4, line 24: Norton et al. (2008) is not in the reference list.

Page 4, line 25: it is laudable that the authors try to take snow shielding into account in their calculation, but how reasonable is it to extrapolate a snow-depth – elevation relationship determined for central Switzerland to eastern Austria? The most comprehensive climatology database to date that I know of (Frei and Schaer, Int. J. Clim. 1998) shows that both mean-annual and winter precipitation is significantly lower in eastern Austria than in central Switzerland.

Page 4, lines 25-27: Can you provide some information on the geology of the sampled catchments, at least reporting the aerial percentage of quartz-bearing lithologies in

Table 2? Were topographic and relief measures only calculated on the quartz-bearing part of the catchments or the entire catchments?

Page 5, line 25: the Roering et al. (2001) model was actually designed to model shallow landsliding, not really hillslope creep.

Page 6, line 1: "higher erosion rates in general" is unclear: what erosion rates are you discussing here?

Page 6, lines 5-9: it could be useful here to show a plot of the combined datasets (the current dataset and that of Legrain et al., 2015).

Page 6, line 25: "segmented" rather than "segmenting".

Page 7, line 1: A similar relief structure was described for glacially influenced catchments in the western Alps by van der Beek and Bourbon (Geomorphology, 2008).

Page 7, lines 11-12: The Vernon et al. (2009) reference is inappropriate here, as these authors did not discuss frost-cracking as a potential mechanism controlling spatial variations in erosion rates (moreover, these authors were looking at long-term exhumation rates from thermochronology data, on which the influence of frost cracking would be much harder to substantiate).

Page 7, lines 12-16: these arguments to rule out frost cracking as a mechanism controlling the variation of erosion rates are not completely convincing. First, it would be good to show the correlation between mean catchment elevation and erosion rate and to show that this correlation is weaker than that between mean catchment slope and erosion rate (this is what the authors appear to argue). Second, the fact that basins of the same average elevation show large differences in erosion rate does not necessarily rule out frost cracking, as this process depends on mean-annual temperature and its variation rather than elevation (which is just taken as a convenient proxy). The aspect of the basins (north- versus south-facing) as well as their geology may play a major role in modulating frost-cracking efficiency.

Page 7, line 30: slab detachment has become the preferred "deus-ex-machina" mechanism to "explain" uplift rates in the Alps. The data reported by Qorbani et al. (2015) provide only a very indirect indication for possible slab detachment. In the absence of more clearly resolved seismic tomography imagery for the European Alps, I feel we should be careful in invoking this mechanism . . .

Page 8, lines 8-15 (and Figure 6). There is something in this Figure I do not understand. Apparently (unless there is a problem with the x-axis) this slope-area plot is for extremely small catchment areas (<1 km2), i.e. for the most part within the hillslope domain (the hillslope – fluvial transition typically occurring at catchment areas of 105-106 m2. At these small catchment areas, the data should show either increasing slope with area (diffusional hillslopes) or no relationship between slope and catchment area (landslides, debris-flow domain). Yet the data show very good slope-area scaling, with larger concavities for glaciated than for non-glaciates catchments (as expected). So either the area axis is in km2 rather than m2 (which would make sense) or something very curious is going on.

A more general comment on Section 3.4: possibly your best potential argument for a control by glacial preconditioning on erosion rates would come from your 5 catchments in the Seckauer Tauern. There are 3 unglaciated and 2 glaciated catchments, with for the rest fairly similar characteristics (at first glance at least). There are also 2 catchments that have significantly higher erosion rates than the other 3. Are these the two formerly glaciated catchments? If so, bingo!

General comment on Section 3.5: the fact that erosion rates appear to systematically increase toward the west is, however, not easily explained by a mechanism of glacial preconditioning of topography. On page 9 (lines 11-28) the authors attempt to invoke paleo-climate variations and possibly thicker ice cover in the western Alps, but in the absence of any data this remains somewhat speculative. Several studies have reported average LGM ice thickness for the studied catchment areas; it may be interesting to have a closer look at this, compile this data where it is missing and see if there is a

relationship with millennial erosion rates. However, a more simple relationship may exist between present-day rock uplift (as inferred from GPS studies) and erosion rate – GPS-derived rock-uplift rate data have now been published for most of the Alps, including the western Alps (cf. Nocquet et al., Scientific Reports 2016). If a strong relationship with rock-uplift rates exists (and uplift rates are similar to or higher than erosion rates) then a tectonic or geodynamic control on these laterally varying rates should be invoked.

Page 9, line 4: the arguments used by Persaud and Pfiffner (2004) to suggest active ongoing tectonics in the part of the central Alps they were studying were not particularly convincing. Not sure it is worth citing this here.

Page 9, lines 23-25. This is a long and complex phrase. It is important for your arguments though; you may want to reformulate it.

Comments on Figures

Overall, I'm not sure the organisation of the figures is the most logical and effective. Some could be merged; others appear to be missing.

Figure 1: Needs to show the different regions sampled (Styrian Alps, Levanttal Alps, Gleinalpe, Koralpe, Schladmig Tauern, Seckauer Tauern, Pohorje). A simple way to do this would be to color-code the catchments and add a legend (in that case Fig. 2a would not be needed anymore). An additional panel with a simplified geological map could also be useful here.

Figure 2: (a) can be combined with Figure 1. If you want to keep a map with the catchments here, it may be more useful to color-code them according to rate, so that the reader can see the spatial variation in erosion rates easily. An additional plot of erosion rate as a function of mean-catchment elevation would be useful (see specific comments above).

Figure 6: Check the scale for this plot (see comment above)!

**ESurfD**
Figure 7. Not sure the erosion rate versus slope plot is the most effective here. An interesting plot could be simply erosion rates versus longitude (to show whether there is really an east-west increase or this is only apparent); otherwise suggested plots (see above) would be erosion rate versus average-LGM ice thickness and/or erosion rate versus present-day rock uplift rate (GPS data).

Grenoble, 7 July 2016 Peter van der Beek

Please also note the supplement to this comment:
http://www.earth-surf-dynam-discuss.net/esurf-2016-29/esurf-2016-29-RC2-supplement.pdf

---

## Author Comment (AC1) · 2 Nov 2016

Response to comments

esurf-2016-29 Manuscript

"Glaciation's topographic control on Holocene erosion at the eastern edge of the Alps

By Jean L. Dixon, F. von Blanckenburg, Kurt Stüwe, and Marcus Christl

**Marco G. Jorge**

*I find the manuscript very interesting and look forward to its final version. Below, please find some comments and proposed edits that may help improving the manuscript.*

Thank you to Marco Jorge for his detailed comments that help improve the clarity of our manuscript.

*-- Introduction --*

*Page 2, Sentence L3-5. Suggest rephrasing. Second clause does not preclude first clause.*

Reworded.

*L7. Perhaps, the transition to glacial buzzsaw concept is not properly backgrounded by previous sentence.*

We have reworded this text to clarify.

*L13. Suggest rephrasing for flow with previous sentence. E.g., first say that glacial landscapes can be preserved . . ., influencing. . .*

To transition to the next sentence, we have changed to "Glacial processes significantly alter landscapes, and therefore leave a lasting topographic legacy that …"

*L15. Widely glaciated mountain belts?*

We have changed to "in modern mid- and high-latitude mountain belts".

*L16. References missing.*

This sentence refers to the previous paragraph, which is well cited.

*L18. Rephrase: not all glaciers are. . .*

Changed

*L25. Complete "Climate's influence on . . .*

"Climate's influence on mountain belt erosion"

*—- Section 2.3, Digital Terrain Analysis —-*

*Page 4, L31. Suggest mentioning that SRTM is 1 arc-second, instead of 80 m grid (dependence of cell size on latitude).*

Thank you. We have clarified this point in the text.

*Page 5, L4-5. This remark, specifically the last sentence, is too precise, or off what would be reasonable to discuss based solely on the differences in the topographic metrics. Basin slope gradient average is not indicative of local scale morphometry independently of DEM cell size and an average gives no information about spatial variation. Either remove or rephrase and extend this discussion by adding more information (references?).*

Grid-scale influence on mean slope values have been previously recognized and discussed by a number of authors. Here, we provide a reference to Zhang and Montgomery (1994).

*(Table 2: refer that slope gradient values are averages)*

Done

*Which parameters were used for stream and catchment delineation? Was a flow accumulation threshold used? I presume from Fig 1 that delineation was based on the location of sampling sites (basin outlet). I believe it is important to justify location of sampling sites as well as its influence on basin delineation and morphometry.*

We clarify that catchments were delineated upstream of sample points.

*—- Section 3, results and discussion —-*

*The message would be clearer and the manuscript a better read if the discussion was separated from the results; there would be less back and forth. The discussion would benefit from the inclusion of further morphometrics (e.g., of elevation dispersion). In instances, the conclusions within the discussion overshoot what would be reasonable to conclude from the presented data (see below). I think that it is important to include an evaluation of lithology as a conditioning factor of the observed differences in erosion rates (even if it is null).*

We have separated Results and Discussion sections, and agree that this change makes a significant improvement to the readability and clarity of the manuscript.

*Page 5, L11. which lowland basin catchments? In the Styrian Basin?*

Clarified.

*L22. Rephrase. "these data" refers to both mean elevation and slope gradient but this sentence and following sentences address slope only.*

We have specified that we are referring to the DEM scale.

*L23. Whereby —> where*

Changed to, 'such that'

*L23, 24. The described relationship between mean slope and erosion rates does not imply non-linear relationship. Perhaps reword results.*

This text has significantly changed

*Page 6, L11-13 (paragraph's last sentence) Recommend rephrasing. Remove first clause and reword last clause.*

Thank you. This sentence was unclear. We have reworded.

*L16-21. This is too simplistic. For example, note that glaciated catchments generally are higher in elevation and non-glaciated catchments vary widely in mean elevation (Table 2) (differences in potential energy).*

We have noted this wide variation in mean elevation.

*L25. segmenting –> segmented*

Changed.

*L23-36. Too simplistic and somewhat confusing. Differences in average elevation between basins and elevation-slope relationships within basins are different things. Why should the relative location of the steepest slopes be positively related to basin average slope gradient? Justification for interpreting that to be signal of past glacial sculpting is insufficient.*

The distinct patterns of local mean slopes with binned elevation persist regardless of whether you compare the same elevations between basins, or the distribution of slope within some nondimensional value of relief within the basin.

*Page 7, L11. 'However' should be preceded by semi-colon.*

Changed.

*L14-16. Is the abundance of slopes >35∘ in gradient a good proxy for frost cracking? Address it directly.*

We have reworded this section.

*L24, 25 (Sentence). Explain; and what are area-normalized stream gradients? (area of what?)*

We have significantly improved our description of these figures and metrics.

*L26, 27 (Sentence). Add reference. Last paragraph. What is the authors' take on this discussion?*

We cite Legrain in the previous sentence, and here indicate that their results are only applicable to non-glaciated portions of the Eastern Alps.

*Page 8, L5-7. It was referred before that Legrain et al., 2015 looked at non-glaciated basins. Does "previously suggested" refer to Legrain et al., 2015?*

Changed.

*—- Conclusions —- Page 9, L1. "Repeated" meaning supporting previous studies? –> Add references*

Here, we refer to multiple sets of data presented in this study, not to previous work.

*Page 10, L2, 3. Not clear where these values are from; add references?*

Data presented without references are from this study. We cite Legrain et al., 2015 for their background erosion rates.

---

## Author Comment (AC2) · 2 Nov 2016

Response to reviewer comments

esurf-2016-29 Manuscript

"Glaciation's topographic control on Holocene erosion at the eastern edge of the Alps

By Jean L. Dixon, F. von Blanckenburg, Kurt Stüwe, and Marcus Christl

*Pierre Valla – Reviewer 1*

Thank you to Pierre Valla for his complete and insightful review. Below we detail how we address the reviewer's comments.

*This is a very interesting, well-written and presented manuscript. The authors have really well introduced their work with a comprehensive review of the literature and presentation of the open questions concerning the late evolution of the European Alps. They propose new detrital 10Be-derived erosion rates that nicely complement previous observations in the Eastern Alps and have performed a detailed topographic analysis to discuss their dataset and the spatial variability of catchment-wide erosion rates over the entire European Alps. As such this study represents in my opinion a valuable contribution and will have potential of great interest for the community. I have some questions and suggestions, mainly to better clarify some of the results and their interpretation, and have outlined them in a set of general and specific comments below.*

We thank the reviewer for this assessment of the manuscript.

*General Comment 1: Structure: the present manuscript is overall well-structured, with clear "Introduction" and "Approach" sections. However, the merged section 3 "Results and Discussion" is sometimes quite difficult to follow between the different sub- sections. I would suggest the authors to separate the results from the discussion which may be easier to follow for the readers.*

We have separated the results and discussion into two sections in our revision. We believe this edited structure helps improve the logical flow of the manuscript.

*General Comment 2 - 10Be-derived basin erosion rates: the new results for 26 basins presented by the authors are really interesting and nicely complement previous investigations of Holocene erosion rates across the European Alps. However, I would recommend the authors to provide more discussion about the possible bias in calculated erosion rates from local complexities and the resulting implications for their story. I have listed some detailed questions below, especially concerning the integration times, snow-cover correction, the sediment grainsize, the approach for floodplains and non-quartz bearing areas... Moreover, some discussion about the (previously) glaciated catchments would be helpful for ESurf readers concerning the re-mobilization of morainic/glaciogenic material, or the glacial perturbation on 10Be concentrations which might be non-negligible for slowly-eroding terrains (e.g. Glotzbach et al., 2014 Terra Nova). Finally, the compilation of existing 10Be-derived basin erosion rates from the literature is very interesting and nicely put the new dataset into a broader context. I am wondering if 10Be-derived basin erosion rates have been directly taken from published papers, or if the authors have recalculated the erosion rates with a uniform/updated production rate? If not, maybe discuss also this point and potential implications.*

The reviewer points out several important points that we clarify in the revised manuscript. We made several assumptions regarding erosion rates in the glaciated basins that we now better address.

A) Glacial perturbation on [10]Be concentrations

The reviewer is correct that the glacial history of several of our catchments may influence cosmogenically derived erosion rates in ways we do not address. Firstly, glacial perturbations may result in non-trivial differences between erosion rates calculated assuming either steady or non-steady state 10Be concentrations, especially in slowly eroding terrain (e.g., Glotzbach et al., 2014). We now compare erosion rates derived from steady state assumptions to those derived using non-steady state calculations from Lal 2001. In the case of our glaciated catchments, with erosion rates between 170-230 mm/ky, assuming 10Be concentrations have reached steady state may result in the overestimation of erosion rates by 9%. This assumption results in a non-trivial, but still relatively small bias to calculated erosion rates considering our glaciated basins erode roughly a factor of two times faster than non-glaciated basins, and up to a factor of five times faster than background erosion rates near 40 mm/ky. We address this assumption and bias in the revised manuscript, and additionally reference Glotzbach et al. (2014), Wittmann et al. (2007) and Norton et al. (2010) who have previously addressed this issue.

The reviewer also requests that we address other impacts of glaciation on [10]Be signals, such as the re-mobilization of glaciogenic material. Though we do not have data that speak directly to this issue, we expand our discussion of results in the revised manuscript to more broadly discuss the varied complications that arise using [10]Be concentrations in previously glaciated terrain. Wittmann et al. (2007) and Delunel et al. (2014) suggest glaciogenic sediment may contain inherited [10]Be concentrations if glaciers have incompletely zeroed surface concentrations via shallow erosion or if glacial advance overrode soils and later incorporated them into glacially eroded sediments. In this case, [10]Be concentrations may instead underestimate erosion rates, though this effect should be the largest in currently glaciated or recently glaciated catchments. Furthermore, Wittmann et al. (2016) measured [10]Be concentrations upstream and downstream of several glacial features that may influence sediment storage: deep lakes formed during glacial retreat and large moraine amphitheaters). These authors found negligible effects of sediment storage on [10]Be concentrations.

B) Compilation of erosion rates

In the submitted manuscript, figure 7 shows erosion rates previously published across the Alpine range. In this original compilation, we did not recalculate erosion rates using a uniform or updated production rate. Our intent was to show the range and variability of erosion rates, and not to publish a comprehensive compilation of erosion rates. However, considering that differences in assumed production rates between published studies may alter the data shown in this figure, in our revised manuscript we recalculate erosion rates to a consistent sea level, high latitude production rate of 4 at/g/yr, regardless of the original scaling factors used. This rescaling of published erosion rates to a consistent production rate does not significantly alter this figure, nor change interpretation of the data across the Alps. However, it does increase the usefulness of our compilation and improve comparison of rates across disparate studies.

*General Comment 3- Topographic metrics: the authors provide a detailed topographic analysis of the studied catchments with a special attention on mean basin slopes and slope distributions within the basins. I would suggest to also report other metrics, such as hypsometry or local relief to strengthen their message (see also specific comments below).*

We have added measurements of mean local relief into the manuscript, and the data is now provided in table 2.

*Finally, the authors discussed in some sections the role of fluvial incision and river response to potential perturbations, but this would be better illustrated with some river profiles shown as a new figure. This will clearly help readers to evaluate the degree of disequilibrium in river profiles, and potential differences between regions/massifs or glaciated vs. non-glaciated catchments.*

We choose to focus our analyses on hillslope processes within catchments rather than fluvial. The manuscripts by Legrain et al. (2015) and Robl et al. (2008) have previously presented and discussed detailed river profiles across our study region of the Austrian Alps.  However, our figure 6 does provide accumulation area scaling of local slope gradients and allows us to compare how hillslope gradients and channel slopes both vary across our catchments.

*Specific comments, by line number: - Page 1, lines 26 and 27: "poor topographic indicators of controls" (l.26) contradicts with the end of the next sentence "its topographic legacy" (l.27). Maybe rephrase the sentence on line 26 (which also appears in contradiction with the manuscript's title).*

This text has been deleted.

*Page 2, lines 14-15: "suggest that glacial forcings are the dominant control on land- scape evolution in mountain belts". This sentence reads vague and quite general, maybe rephrase to precise in which mountain belts (mid- and high-latitudes?) and over which timescales (Plio-Quaternary?).*

This sentence now specifically references "modern mid- and high-latitude mountain belts".

*Page 2, line 26: "have been invoked as principle drivers of erosion and uplift". Maybe cite also there Fox et al. 2015 (Geology).*

Citation added.

*Page 2, line 28: "post-glacial erosion may explain rates of uplift in the region via isostasy". Please clarify the timescales over which this correlation is valid, i.e. Holocene for erosion rates from Wittmann et al. (2007) and historical/modern for uplift rates (leveling data from Schlatter et al., 2005).*

This section now explicitly notes that Wittman compared millennial-scale erosion rates to modern rates of uplift.

*Page 2, line 29: "youthful tectonic features such as river knickpoints". Please rephrase, river knickpoints can have several origins and as rightly stated after ("correlated with previous glacial cover...") these are not tectonic markers.*

We have removed the interpretation of 'youthful tectonic features' and simply note that Norton et al (2010) correlated knickpoints to glacial extent.

*Page 3, line 8: "erosion rates". Please specify the timescale "post-LGM/Holocene" for these erosion rates.*

Changed.

*Page 3, lines 13-14: Please indicate the massifs, basins and other important locations on Figure 1 for clarity.*

We have labeled more locations in the figure for context.

*Page 3, line 17: "that has uplifted some 300 m above sea level in the last 7 Ma". Please add a reference here if possible.*

Reference added to Legrain et al., 2014.

*Page 3, line 18: "Massifs" corrected by "massifs".*

Corrected

*Page 3, line 27: "This timing coincides with inversion and uplift of the Styrian and...". Please clarify which timing is considered here. Is it 4 Ma (l.27), 7 Ma (l.17) or 10 Ma (l.16)?*

This passage immediately follows the sentence referencing the initiation of incision at 4 Ma. We have clarified this point by changing this text to read, 'The timing of incision...'.

*Page 4, line 9: "250-500 um size fraction". In Table 1, some sample sizes are higher (500-800 um). Are they replicates? Please clarify, and maybe also discuss potential implications when comparing the resulting erosion rates for different grainsizes.*

Our [10]Be analyses were run on the find sand (250-500 um) size fraction. However, three samples were sieved at both 250-500 um and 500-800 um size fractions to check for grain size dependence of [10]Be analyses. We clarify this in the text.

*Page 4, line 25: "elevation-snow depth relationships previously determined in the Swiss Alps". I am not sure to what extent the data from Auer (2003) in the Swiss Alps can be extrapolated to the Austrian/Slovenian Alps (this extrapolation might depends on the local precipitations and moisture patterns). How comparable are the two climatic settings? Please discuss this extrapolation and the potential implications for calculated erosion rates.*

Snow shielding, calculated here, represents an 8% correction to production rates, and therefore is a non-trivial calculation. We recognize that elevation-snow depth relationships determined across the Swiss Alps may not universally apply to our study region in Austria and Slovenia, nor do they likely reflect snow conditions across the millennial timescales over which cosmogenic [10]Be integrates. However, they do provide best-available constraints for snow shielding. This is now clarified in the text, and table 2 provides our data for snow-shielding so the reader can assess the magnitude of this correction.

*Page 4, line 26: "we set production rates equal to zero in parts of drainage basins with carbonate terrains". These areas should be excluded from the calculations (since they do not contribute quartz) and not set to zero-production, otherwise this would introduce a bias in the catchment-wide erosion rates. Please correct or clarify. Also, how did the authors consider low-gradient areas such as floodplains (figure 1) in their calculations? Are they included in the integrated 10Be erosion rate calculations? Please clarify.*

Thank you for pointing out this unclear text. Carbonate terrain are excluded from calculations, and this is clarified now in the text. We did not specifically consider floodplains in our calculations, and all quartz-bearing portions of the catchment are included in 10Be calculations and topographic metrics. Floodplains do not make up significant portions of our basins, primarily because even though catchment size is typically over 50 km², the majority of our catchments are hilly to mountainous. Interestingly, glaciated catchments may contain high altitude, low gradient areas such as cirque valleys. If these portions of the landscape did not deliver sediment, perhaps because trapped in cirque lakes, then they should also be removed from calculating production rates. Hence catchment-wide production rates would decrease, and so would denudation rates. This could result in erosion rates in glacially conditioned catchments to be lower than calculated. We now discuss this complication in section 3.3.

*Page 5, line 14. The (previously) glacial setting for these catchments may imply potential biases in the 10Be concentrations and thus in the calculated erosion rates, with input from morainic/glaciogenic material (e.g. Delunel et al., 2014 ESPL) or the impact of former glaciation (Glotzbach et al., 2014 Terra Nova). I would suggest the authors to discuss these points further and to what extent they may perturb the inferred catchment-wide erosion rates.*

Please see our response to General Comment 2 above.

*Page 5, line 21: "Measured erosion rates also generally increase with increasing mean basin elevation". Can the authors provide a figure for this correlation?*

This text is now removed; however, elevation data is provided in Table 2.

*Page 5, line 22: "Measured erosion rates also generally increase with [...] slope". Looking at Figure 2b, we can also see two clusters: 1) non-glaciated basins with no correlation between slope and erosion, and 2) (previously) glaciated basins where there seem to be an inverse correlation between slope and erosion. Please consider discussing this potential alternative observation.*

We now explicitly address this point in the second paragraph of section 4.1.

*Page 5, line 28: "We note that both erosion rates and catchment mean slope correlate with the proportion of the catchment that exceeds 35° (Fig. 2b)". This is not shown on figure 2b, or maybe I missed something. Please correct or clarify.*

This line mistakenly referenced a figure we had removed prior to submission. As stated on the response to Page 5, line 21, we have expanded Figure 3 to add this plot.

*Page 6, lines 9-14: "therefore likely reflect this erosional response to river incision and tectonic processes across the range". In the previous section (3.1, lines 11-12), the spatial variability in erosion rates was suggested to reflect lithological variations. Please clarify. –*

The previous section was describing the catchments, not attributing controls on erosion rates. However, we have better clarified this text by indicating that Legrain found "Higher rates in the Styrian Basin compared to uplands of Koralpe, therefore likely reflect this erosional response to river incision and tectonic processes across the range, rather than lithologic differences."

*Page 6, line 25: "We segmenting". Please correct.*

Corrected

*Page 7, line 1: "consistent with characteristic glacial and non-glacial slope-elevation curves predicted by Robl et al. (2015)." This specific slope distribution with elevation for glaciated terrains has already been observed in other places (e.g. van der Beek and Bourbon, 2008 Geomorphology). Please cite some references.*

Thank you for pointing out this very relevant paper. We have now included this reference in several places of our revised manuscript.

*Page 7, line 7: "other climatic controls such as precipitation rates". How variable are the mean precipitation rates between the different studied catchments? Please discuss this point.*

We have added text to address this point at the bottom of this paragraph. We now state, "Furthermore, mean annual precipitation is likely a poor indicator of erosion in our unglaciated catchments since areas of the Mürz valley that display the highest non-glacial erosion rates tend to be drier than more slowly eroding portions of the Koralpe range (~BMLFUW, 2007)."

*Page 7, line 11: "correlations between elevation and either rock uplift or erosion rates". Unclear, please clarify. I think this would rather be "correlations between elevation or rock uplift and erosion rates". Also,*

*correlations from Vernon et al. 2009 are based on thermochronology and thus imply much longer timescales than Holocene, and they do not consider frost-cracking. Please correct.*

We have changed this to read "correlations between elevation and erosion rates." The Vernon reference has been removed.

*Page 7, line 13: "elevation poorly correlates with the abundance of steep (>35ÌŁ) slopes". On Figure 3, there seem to be some non-linear correlation between elevation and fraction of the basin >35˚. Please rephrase or discuss further this point.*

This section now reads, "elevation poorly correlates with the fraction of steep (>35˚) slopes, notably in the rapidly eroding, previously-glaciated basins where the abundance of steep topography varies widely despite similar mean basin elevations".

*Page 7, line 16: "While frost-cracking may enhance erosion at alpine sites, it does not appear to explain the patterns and variability in erosion rates across our catchments". Maybe frost-cracking is occurring (or had occurred in the Lateglacial period) for previously-glaciated catchments, but for non-glaciated catchments the mean elevations are too low to consider this effect. Would it be possible?*

The reviewer notes an interesting potential correlation between the intensity of frost cracking and past glacial history, while we are not able to completely rule out this mechanism, we have expanded our discussion to provide multiple lines of evidence that it is not the primary control. The text now reads:

"It might be hypothesized that the intensity of frost-cracking processes are (or were) greatest in our previously glaciated catchments, thus potentially explaining the distribution of erosion rates. Across our study basins, catchment mean slope and elevation are correlated (Fig. 3), however, elevation poorly correlates with the fraction of steep (>35˚) slopes, notably in the rapidly eroding, previously-glaciated basins where the abundance of steep topography varies widely despite similar mean basin elevations. Therefore, the elevational proxy for frost cracking does not correspond to topographic indicators of rapid erosion in our study area. Furthermore, this mechanism is not supported if elevation is a proxy for the intensity of frost-cracking, as we find large differences in erosion rates at basins of the same elevation (Table 2). While frost-cracking may enhance erosion at alpine sites, it does not appear to explain the patterns and variability in erosion rates across our catchments."

*Page 7, lines 24-25: "If catchment erosion were driven by increased river incision, then we should observe higher area-normalized stream gradients in rapidly eroding catchments". What are "area-normalized stream gradients"? Also, did the authors study the river profiles to identify such perturbations. I would suggest the authors to show some river profiles and/or hypsometric curves for the studied catchments to illustrate the discussion about river incision (see also my general comment). Same question for the sentence on Page 8, line 7-8. This is difficult to see the fluvial domain on figure 6, so I would encourage the authors to show some figures focused on the fluvial part of catchments (river profiles, slope-area diagrams).*

Here, we have clarified this text by rewording as:

"If catchment erosion were driven by increased river incision, then we would expect steeper stream gradients in rapidly eroding catchments. Legrain et al. (2015) observed correlations between higher normalized stream steepness indices and erosion rates within the Koralpe region of our study area, but only within small non-glaciated catchments."

As noted earlier, we focus specifically on hillslope erosion in this manuscript. The manuscripts by Legrain et al. (2015) and Robl et al. (2008) have previously presented and discussed detailed river profiles across our study

region of the Austrian Alps.

*Page 8, lines 3-6. How about lithological variations (and thus erosional resistance) as a potential control on the spatial variability in erosion rates?*

Lithology may indeed influence erosion rates across the range, however not so much as to overprint glacial history. For example, the weakest lithology is found in the Styrian basin catchments, which are underlain by Miocene sediments. Erosion rates in this region are slightly higher than those in the adjacent Koralpe range, underlain primarily by much stronger bedrock. However, these erosion differences are much smaller than those found between previously glaciated and unglaciated basins.

*Page 8, line 15: "processes solely within the hillslope domain". Did the authors also look at local relief as a potential topographic metrics for erosion rates?*

We now provide measurements of local relief in table 2, which scale closely with slope measurements.

*Page 8, line 22: "Compiling previously reported cosmogenic 10Be-derived rates across the Alps". Did the authors report here the original erosion rates or did they recalculate erosion rates with uniform/updated production rate? Please clarify and discuss potential implications for comparing erosion rates across the Alps.*

All previously published rates compiled in Figure 7 and discussed here have been rescaled to a consistent high-latitude, sea-level production rate of 4.0 a/g/y. Please see our related response to general comment B.

*Page 9, line 1: "we might expect Holocene erosion to reflect exhumation and uplift…". How can erosion reflect exhumation and uplift? This appears unclear, please rephrase.*

This text has been changed to, "We might expect Holocene erosion to reflect uplift or rates of long term exhumation across the range."

*Page 9, lines 4-7: "Long-term exhumation rates from thermochronometric ages are largely attributed to deep tectonic processes that increased during the Cenozoic". Please rephrase, long-term exhumation rates are also driven by Plio-Pleistocene changes in erosion following climatic forcing as well as drainage modifications, not only tectonics.*

We have rephrased to state that these rates have been partially attributed to deep tectonic processes, thus not leaving out other interpretations.

*Page 9, lines 12-13: "with highest modern and LGM precipitation occurring in the northern slopes of the Alps and decreasing to the south and east". Moisture patterns have changed between the LGM and modern (Florineth and Schluchter, 1998) so I am not sure that precipitation maxima have always been on the northern Alpine slopes. Please clarify.*

The citation has been altered to (Florineth and Schlüchter, 2000). Though the strength of in the westerly and southerly airflows have altered precipitation patterns, these authors suggest that generally both modern and LGM climates suggest decreasing precipitation to the east.

*Page 10, line 3: Please correct "mm/ka" by "mm/ky" for consistency.*

Changed.

*Page 10, line 5: "49 mm/yr" would rather be "49 mm/ky". Please correct.*

Changed

*Tables and Figures: - Figure 1: I would suggest to add main massifs, basins and maybe river names on Figure 1b to help the readers following section 2.1 and to link with subsequent figures.*

Done

*Figure 2: Please indicate replicates on Figure 2b with a star or different symbol. Also, I would suggest to also add the data from Legrain et al. (2015) on this figure, that may be helpful to compare them already at this stage, no (as they appear on figure 3)?*

Figure 2 and 3 only show data generated using the 250-500 um size fraction (i.e., no replicates are shown). Table 2 provides data for other size fractions for comparison.

*Figure 3: Maybe use different symbols (or open/filled) to differentiate between glaciated/non-glaciated basins?*

We have distinguished previously glaciated and unglaciated basins in Figure 2. We believe adding new symbols to Figure 3 would make the figure too busy and decrease readability.

*Figure 4: Why is the "basin erosion" legend reversed in panel a? Panels c and d are nice and informative, I am wondering if similar panels with elevations would be informative? On panel b, what are the criteria for "partially glaciated" and how does it relate to figure 2 with glaciated/non-glaciated? Please clarify.*

The legend and color scale for basin erosion are consistent across all figures. We have added text to the results section and legend to figure 2 to clarify partially glaciated basins, in which only the uppermost elevations were glaciated.

*Figure 5: Would it be possible to use different symbols for glaciated/non-glaciated catchments?*

Please see our response to the comment on Figure 3.

*Figure 6: This figure is difficult to read at present. I think that the slope distribution for the hillslope domain is already illustrated by panels c and d of figure 3, so I would recommend to show here only the fluvial domain (>103 m2) to better highlight any differences between the different rivers.*

Figure 6 shows how slope distributions scale with accumulation area, and therefore highlight both the hillslope (far left) and fluvial (far right) domain.

*Figure 7: I would be curious to see if there is any correlation between mean basin elevation and erosion rate across the Alps. Did the authors look at this or can add the corresponding figure if informative?*

Though we do not provide this data in a figure, it is available in table 2. However, erosion rates are poorly correlated with elevation across unglaciated basins.

*Table 1: I would suggest to also indicate the "integration time" for the reported denudation rates. In the footnote, please correct "negative and fast muons" by "slow and fast muons".*

*Table 2: In the footnote, please replace "Pleistocene" by "LGM". What is the maximum ice coverage during the Pleistocene (do we have evidence ofr more extended glaciations before the LGM in this part of the Alps)?*

Changed

*I hope these comments and suggestions may be useful for revising the manuscript, and I look forward to seeing it published.*

Many thanks again to this reviewer for helpful comments that have improved our manuscript.

*Peter van der Beek – Reviewer 2*

Many thanks to Peter van der Beek his thoughtful and insightful review. Below we detail how we address the reviewer's comments.

*Dixon et al. provide new detrital cosmogenic 10Be data to constrain erosion rates of nearly 30 catchments in the easternmost Alps (Austria and Slovenia). While earlier studies in this area have argued for a tectonic control on erosion rates, with catchments influenced by recent uplift recording higher rates than catchments to which this recent phase has not (yet) been communicated, the extended dataset presented here shows that the main controlling parameters on erosion rates are basin relief and mean slope, which the authors argue to be influenced by glacial preconditioning.*

*This study provides interesting new data that significantly tone down previous interpretations, and provides an integrated view of Holocene erosion rates in the Alps. It is therefore timely and definitely suitable for publication in Earth Surface Dynamics. While my overall evaluation of this manuscript is thus positive, I recommend it be returned to the authors for moderate revisions before final acceptance. These pertain to some apparent misconceptions or imprecisions in the writing, as well as the intriguing slope-area relationships that may merit some more discussion. As most of my comments are rather specific, I will list them tied to page and line numbers below:*

*Page 1, line 13: the Hergarten et al. model is based on a fundamental misconception: it mistakes a glacial imprint on topography for a transient tectonic signal. It would be preferable if this fundamentally flawed study were not perpetuated in the literature any more than it needs to; I would thus suggest the authors to refrain from citing it, particularly in the abstract.*

We have removed citations from the abstract, but not from the remaining text. Regarding the Hergarten et al. paper, we recognize that it presents an alternative, and to some controversial, view to common Alps geomorphologic paradigms. In this manuscript, we cite the paper primarily for its role in the debate on tectonic vs glacial controls on topography and erosion – regardless of whether its view is widely accepted. As for the controversy about the causes for the lack of steep slopes at high elevations in the Alps we note that the jury is still out on this matter and a whole sale rebuttal of the Hergarten model is unjustified. In particular we note that (a) large parts of the eastern Alps is characterized by karstified plateaux bearing Oligocene fluvial gravels indicating that they cannot be caused by glacial processes (Frisch et al., 2000); (b) planation surfaces are also found on peaks above 2000 m surface elevation that are outside the glacial icecap (Legrain et al., 2014); and (c) Morphometric analysis of the Alps shows that glaciated and never glaciated parts of the Alps show similar slope-elevation distributions (Robl et al., 2015).

*Page 1, line 23: Although Legrain et al. do invoke "deep lithospheric processes", it is not sure these are required for the easternmost Alps. In contrast to the west, convergence is still active in the East (e.g. Serpelloni et al., Geophys. J. Int., 2005) and the inversion of the Pannonian basin can be linked to a change*

*in crustal stress fields from extension to compression (itself possibly linked to a deep lithospheric cause, however).*

We have removed the citation from the abstract.

*Page 2, line 7. The process of valley deepening and widening described above is not, in fact, the cause of the "glacial buzzsaw". The generation of widespread low-relief surfaces at elevations around the average Quaternary ELA (the topographic fingerprint of the "buzzsaw") is rather linked to efficient cirque retreat, possibly aided by periglacial (frost-cracking) processes. See Mitchell and Montgomery (Quat. Res., 2006) and Egholm et al. (ESurf., 2015) for discussions of these processes.*

Thank you for clarifying the process behind the term. We edited this text to address the buzzsaw in general terms and not link to valley widening and deepening.

*Page 2, line 9: Isostatic rebound will cause rock uplift but will not in itself increase relief. Relief increase is due to the fact that glacial erosion is strongly non-uniform or "selective", deepening valleys while having limited effects on higher parts of the landscape.*

This section is reworded, "Glacial erosion may increase mountain relief and cause isostatic uplift of rocks."

*Page 2, line 14: Norton et al. (Geology, 2010) would be a good complementary (or alternative) reference here.*

We reference this work by Norton elsewhere in the manuscript, but do not find it more relevant than current citations at this text.

*Page 2, lines 26-27: This presentation of the findings of Wittmann et al. (2007) is slightly misleading. In fact, their regression of denudation rates versus rock-uplift rates gave a slope of 1.0±0.25, i.e. erosion rates could be either higher or lower than rock- uplift rates, and these authors include a lengthy discussion of the potential implications of this finding. Champagnac et al. (2009) did subsequently argue, based on a subset of this data, that rock-uplift rates were lower than denudation rates, but even their analysis is not equivocal on this point.*

We have changed this text to state: "Wittmann et al. (2007) and Champagnac et al. (2009) noted that erosion rates scale with – and may exceed – uplift rates in the central Alps,

*Page 3, line 1: again, why do you need to invoke deep lithospheric processes in a region where convergence is still ongoing?*

We have removed this text as the underlying mechanism is unimportant in this section. The text now reads, "In the eastern portion of the range, accelerated rates of river incision and hillslope erosion since 5 Ma have been suggested to record late Tertiary uplift (Legrain et al., 2015; Wagner et al., 2010)."

*Page 3, lines 12-14: there are several regional names here (Styrian (Alps?), Levanttal Alps, Gleinalpe, Koralpe, Schladmig Tauern, Seckauer Tauern, Pohorje) that are not know to a non-Austrian readership. They should be indicated on the map of Fig. 1.*

We now cite figure 1 earlier, provide labels on figure 1, and reference figure 2 (which further highlights these regions).

*Page 3, lines 16-18: a geological map might make this description of the regional geology easier to follow.*

Though we recognize the use of an extra site figure, we have elected not to include a geological map as our study sites span two separate countries with lithologic boundaries not easily merged as mapped.

*Page 3, lines 27-28: "but appears conspicuously unrelated" to what? This is unclear*

This text is deleted.

*Page 4, line 24: Norton et al. (2008) is not in the reference list.*

Reference added.

*Page 4, line 25: it is laudable that the authors try to take snow shielding into account in their calculation, but how reasonable is it to extrapolate a snow-depth – elevation relationship determined for central Switzerland to eastern Austria? The most comprehensive climatology database to date that I know of (Frei and Schaer, Int. J. Clim. 1998) shows that both mean-annual and winter precipitation is significantly lower in eastern Austria than in central Switzerland.*

Please see our detailed response to Reviewer 1 regarding snow shielding.

*Page 4, lines 25-27: Can you provide some information on the geology of the sampled catchments, at least reporting the aerial percentage of quartz-bearing lithologies in Table 2? Were topographic and relief measures only calculated on the quartz-bearing part of the catchments or the entire catchments?*

*Page 5, line 25: the Roering et al. (2001) model was actually designed to model shallow landsliding, not really hillslope creep.*

This text has been changed to "or by non-linear diffusive transport"

*Page 6, line 1: "higher erosion rates in general" is unclear: what erosion rates are you discussing here?*

This sentence fragment has been removed.

*Page 6, lines 5-9: it could be useful here to show a plot of the combined datasets (the current dataset and that of Legrain et al., 2015).*

We have added Legrain data to expanded panels in Figure 2.

*Page 6, line 25: "segmented" rather than "segmenting".*

This sentence is reworded.

*Page 7, line 1: A similar relief structure was described for glacially influenced catch- ments in the western Alps by van der Beek and Bourbon (Geomorphology, 2008).*

Citation added.

*Page 7, lines 11-12: The Vernon et al. (2009) reference is inappropriate here, as these authors did not discuss frost-cracking as a potential mechanism controlling spatial variations in erosion rates (moreover, these authors were looking at long-term exhumation rates from thermochronology data, on which the influence of frost cracking would be much harder to substantiate).*

Thank you. The citation has been removed.

*Page 7, lines 12-16: these arguments to rule out frost cracking as a mechanism controlling the variation of erosion rates are not completely convincing. First, it would be good to show the correlation between mean catchment elevation and erosion rate and to show that this correlation is weaker than that between mean catchment slope and erosion rate (this is what the authors appear to argue). Second, the fact that basins of the same average elevation show large differences in erosion rate does not necessarily rule out frost cracking, as this process depends on mean-annual temperature and its variation rather than elevation (which is just taken as a convenient proxy). The aspect of the basins (north- versus south-facing) as well as their geology may play a major role in modulating frost-cracking efficiency. Page 7, line 30: slab detachment has become the preferred "deus-ex-machina" mech- anism to "explain" uplift rates in the Alps. The data reported by Qorbani et al. (2015) provide only a very indirect indication for possible slab detachment. In the absence of more clearly resolved seismic tomography imagery for the European Alps, I feel we should be careful in invoking this mechanism . . .*

We have reworded this sentence to more carefully cite this related study. We now state:

"seismic anisotropy suggests slab detachment could provide the tectonic mechanism for surface uplift in this Eastern region (Qorbani et al., 2015)."

*Page 8, lines 8-15 (and Figure 6). There is something in this Figure I do not understand. Apparently (unless there is a problem with the x-axis) this slope-area plot is for extremely small catchment areas (<1 km2), i.e. for the most part within the hillslope domain (the hillslope – fluvial transition typically occurring at catchment areas of $10^5$- $10^6$ m$^2$. At these small catchment areas, the data should show either increasing slope with area (diffusional hillslopes) or no relationship between slope and catchment area (landslides, debris-flow domain). Yet the data show very good slope-area scaling, with larger concavities for glaciated than for non-glaciates catchments (as expected). So either the area axis is in km2 rather than m2 (which would make sense) or something very curious is going on.*

Importantly, the reviewer's careful scrutiny of this figure led us to uncover a mistake in how the x-axis was derived and presented. In the original figure, the x-axis was a pixel-based accumulation metric mislabeled as an accumulation area (meaning a value of 100 reflected the upslope area of 100 pixels, not 100 m$^2$ as presented by the axis label). This calculation was performed on a 10 m DEM, and therefore our x-axis scaling was off by two orders of magnitude (such that 100 pixels reflects $10^4$ m$^2$). We have fixed this axis scaling in the revised figure, though we note that this small change does not alter the interpretation nor specifically address the reviewer's confusion above. As the reviewer points out, the data shown in this figure resides primarily in the hillslope domain. The expected scaling the reviewer mentions is one observed in log-log space. Our plot provides slope in degrees as a linear y-axis. These slope values are primarily between 5-35 degrees, and therefore far too steep to represent channels, which normally exhibit slopes <1 degree. We indeed observe a strong decrease in mean slope with log accumulation area in this plot, however in a traditional log slope – log area plot, the scaling of this hillslope domain would appear subdued and near horizontal, especially when compared to fluvial slope values that vary across several, very small orders of magnitude).

*A more general comment on Section 3.4: possibly your best potential argument for a control by glacial preconditioning on erosion rates would come from your 5 catchments in the Seckauer Tauern. There are 3 unglaciated and 2 glaciated catchments, with for the rest fairly similar characteristics (at first glance at least). There are also 2 catchments that have significantly higher erosion rates than the other 3. Are these the two formerly glaciated catchments? If so, bingo!*

Indeed the reviewer is correct that these morphometric differences are primarily delineated in the Seckauer Tauern between glaciated and only partially glaciated catchments. We have altered the manuscript to better

point this out, and figure 2 now provides distinct symbols based on glacial history.

*General comment on Section 3.5: the fact that erosion rates appear to systematically increase toward the west is, however, not easily explained by a mechanism of glacial preconditioning of topography. On page 9 (lines 11-28) the authors attempt to invoke paleo-climate variations and possibly thicker ice cover in the western Alps, but in the absence of any data this remains somewhat speculative. Several studies have reported average LGM ice thickness for the studied catchment areas; it may be interesting to have a closer look at this, compile this data where it is missing and see if there is a relationship with millennial erosion rates. However, a more simple relationship may exist between present-day rock uplift (as inferred from GPS studies) and erosion rate*

We have enhanced this discussion by adding a figure showing compiled erosion rates versus longitude (Fig. 7c). Importantly, we find no systematic east to west variation in erosion rates across the range. Instead, average erosion rates are relatively uniform across the western and central alps despite significant local variability, while only rates in the far eastern study region appear significantly different from other regions of the Alps. This figure highlights that local variability exceeds variability at an orogeny scale.

*– GPS-derived rock-uplift rate data have now been published for most of the Alps, including the western Alps (cf. Nocquet et al., Scientific Reports 2016). If a strong relationship with rock-uplift rates exists (and uplift rates are similar to or higher than erosion rates) then a tectonic or geodynamic control on these laterally varying rates should be invoked.*

Unfortunately, similar reliable data is not easily found for the far eastern Alps.

*Page 9, line 4: the arguments used by Persaud and Pfiffner (2004) to suggest active ongoing tectonics in the part of the central Alps they were studying were not particularly convincing. Not sure it is worth citing this here.*

We have not removed the citation, as the debate in the literature is important, instead we have changed the language to "though some dispute this latter mechanism as a driver of modern rock uplift (e.g. Persaud and Pfiffner, 2004)."

*Page 9, lines 23-25. This is a long and complex phrase. It is important for your arguments though; you may want to reformulate it.*

We have reorganized this paragraph to better formulate our argument that multiple lines of evidence indicate paleoclimate has a lasting imprint on topography and post-glacial erosion rates across the range. The original three arguments in this long sentence are now spread through the paragraph with their relevant citations.

*Comments on Figures*

*Overall, I'm not sure the organisation of the figures is the most logical and effective. Some could be merged; others appear to be missing. Figure 1: Needs to show the different regions sampled (Styrian Alps, Levanttal Alps, Gleinalpe, Koralpe, Schladmig Tauern, Seckauer Tauern, Pohorje). A simple way to do this would be to color-code the catchments and add a legend (in that case Fig. 2a would not be needed anymore). An additional panel with a simplified geological map could also be useful here. Figure 2: (a) can be combined with Figure 1. If you want to keep a map with the catchments here, it may be more useful to color-code them according to rate, so that the reader can see the spatial variation in erosion rates easily.*

The suggestions above include minor reorganizations of figures already provided in the manuscript. Figure 1

provides larger context for the manuscript, and the color-coding of sample catchments by region is provided in figure 2 for ease of comparison with the associated plot of erosion rates.  Figure 4 then provides the color-coding of catchments by erosion rates, for ease of comparison with the detailed slope distributions. This current order of figures follows the organization of manuscript text. We also believe the presentation and separation of results and discussion in the revised manuscript will help improve the logical flow of the manuscript.

*An additional plot of erosion rate as a function of mean-catchment elevation would be useful (see specific comments above).*

We include this data in table 2, though we have chosen not to include this specific figure.

*Figure 6: Check the scale for this plot (see comment above)!*

Please see our response to the reviewer's comment to page 8, line 8.

*Figure 7. Not sure the erosion rate versus slope plot is the most effective here. An interesting plot could be simply erosion rates versus longitude (to show whether there is really an east-west increase or this is only apparent); otherwise suggested plots (see above) would be erosion rate versus average-LGM ice thickness and/or erosion rate versus present-day rock uplift rate (GPS data).*

We agree with the reviewer that an added (or substituted) plot of erosion rates vs. LGM ice thickness or GPS-based uplift rates would be a useful addition. However, unfortunately these data are not compileable across the entire range, primarily due to the lack of consistent quantitative data at our sites at the far eastern edge of the Alps. The current plot of erosion rates vs. slope, is one that allows us to place our data with other previously measured erosion rates, in the context of topographic variables that are measurable across the entire Alpine range. We have added a third panel that shows compiled erosion rates vs. longitude, which helps address patterns of erosion variability across the orogen.

---

## Author Comment (AC3) · 2 Nov 2016

The comment was uploaded in the form of a supplement:
http://www.earth-surf-dynam-discuss.net/esurf-2016-29/esurf-2016-29-AC3-supplement.pdf

---

## Author Comment (AC4) · 2 Nov 2016

The comment was uploaded in the form of a supplement:
http://www.earth-surf-dynam-discuss.net/esurf-2016-29/esurf-2016-29-AC4-
supplement.pdf